JCB Journal of Cell Biology

# Rab46 integrates Ca²⁺ and histamine signaling to regulate selective cargo release from Weibel-Palade bodies

Katarina T. Miteva[1]*, Lucia Pedicini[1]*, Lesley A. Wilson[1], Izzy Jayasinghe[2], Raphael G. Slip[1], Katarzyna Marszalek[1], Hannah J. Gaunt[1], Fiona Bartoli[1], Shruthi Deivasigamani[1], Diego Sobradillo[1], David J. Beech[1], and Lynn McKeown[1]

**Endothelial cells selectively release cargo stored in Weibel-Palade bodies (WPBs) to regulate vascular function, but the underlying mechanisms are poorly understood. Here we show that histamine evokes the release of the proinflammatory ligand, P-selectin, while diverting WPBs carrying non-inflammatory cargo away from the plasma membrane to the microtubule organizing center. This differential trafficking is dependent on Rab46 (CRACR2A), a newly identified Ca²⁺-sensing GTPase, which localizes to a subset of P-selectin–negative WPBs. After acute stimulation of the H₁ receptor, GTP-bound Rab46 evokes dynein-dependent retrograde transport of a subset of WPBs along microtubules. Upon continued histamine stimulation, Rab46 senses localized elevations of intracellular calcium and evokes dispersal of microtubule organizing center–clustered WPBs. These data demonstrate for the first time that a Rab GTPase, Rab46, integrates G protein and Ca²⁺ signals to couple on-demand histamine signals to selective WPB trafficking.**

## Introduction

Endothelial cells (ECs) form the inner layer of blood vessels and, in addition to acting as a barrier between the blood and tissues, are responsible for the rapid response to vascular injury (Chesterman, 1988; McGill et al., 1998). The molecules necessary for this response are pre-stored in specialized vesicles called Weibel-Palade bodies (WPBs; Weibel, 2012). WPBs are EC-specific organelles that, during vascular injury, undergo rapid exocytosis and eject von Willebrand factor (vWF) to attract platelets, thereby initiating platelet plug formation and preventing excessive blood loss (Weibel and Palade, 1964; Wagner et al., 1987; Weibel, 2012; Ferraro et al., 2016). WPBs are formed entirely by the generation of vWF multimers, but they also co-store other cargo that contribute to vascular repair by triggering distinct cellular responses. For example, P-selectin released at the EC surface attracts leukocytes to protect against bacterial invasion (Bonfanti et al., 1989; Dole et al., 2005); angiopoietin-2 (angpt2) stimulates EC migration necessary for wound closure (Fiedler et al., 2004; Hakanpaa et al., 2015); and endothelin-1 is a powerful vasoconstrictor that reduces the vessel surface area (Rondaij et al., 2006). Having a rapid emergency response to vascular injury is vital but potentially risky because, in addition to injury, there are other, functionally distinct, stimulants, such as the proinflammatory amine histamine, that promote WPB exocytosis (van Mourik et al., 2002; Rondaij et al., 2006). The fundamental mechanisms by which diverse agonists coordinate intracellular signals to discriminate between cargo-restricted populations of WPBs remain unclear. Recent studies have started to unravel the regulation of WPB trafficking by Rab GTPases (Nightingale et al., 2009; Zografou et al., 2012; Biesemann et al., 2017).

Rab GTPases are a subfamily of monomeric small GTPases of the RAS superfamily that are master regulators of membrane trafficking (de Leeuw et al., 1998; Pfeffer, 2017). Humans express >60 different Rabs that function in protein trafficking pathways, regulating vesicle formation, movement, budding, and fusion. They have in common the ability to bind and hydrolyze GTP, such that when they are GTP-bound, they are active, and when they are GDP-bound, they are inactive (Pfeffer, 2017). Guanine nucleotide exchange factors and GTPase-activating proteins facilitate interconversion by stimulating the release of bound GDP or the hydrolysis of bound GTP (Novick, 2016). Rab GTPases thus act as molecular switches that are localized to distinct organelles and are able to orchestrate vesicular trafficking by recruiting specific effector proteins.

In addition to G-protein signaling, vesicular trafficking depends on mobilization of intracellular Ca²⁺ (Vischer and

[1]Leeds Institute of Cardiovascular and Metabolic Medicine, University of Leeds, Leeds, UK; [2]Faculty of Biological Sciences, University of Leeds, Leeds, UK.

*K.T. Miteva and L. Pedicini contributed equally to this paper; Correspondence to Lynn McKeown: l.mckeown@leeds.ac.uk.

Wollheim, 1998; Zupancic et al., 2002). The mechanisms by which ECs convert seemingly homogeneous Ca$^{2+}$ signals to an agonist-appropriate cellular response is unclear. Several studies have shown that calmodulin, when activated by Ca$^{2+}$, plays a role in Rab activity (Coppola et al., 1999; Park et al., 2002; Zhu et al., 2016). However, the fundamental mechanism by which Rab GTPases distinguish an agonist-induced rise in intracellular Ca$^{2+}$ to elicit a context-dependent cellular response has remained elusive. We have recently described a novel Rab GTPase expressed in ECs (CRACR2A-L, *cracr2a-a*, NM_001144958.1, which we have renamed Rab46 to distinguish from the short non-Rab isoform: CRACR2A-S, *cracr2a-c*, NM_032680.3; Wilson et al., 2015). Rab46 has been shown to have important functions in signaling and secretion at the immunological synapse (Srikanth et al., 2016). We propose that Rab46 may play a role in Ca$^{2+}$-dependent WPB trafficking in ECs.

Here we show that Rab46 is localized to a subset of WPBs. Acute stimulation of ECs with the proinflammatory mediator histamine (but not the prothrombotic agonist thrombin) evoked trafficking of a subpopulation of WPBs to the microtubule organizing center (MTOC). This retrograde trafficking is dependent on the nucleotide-bound form of Rab46 and dynein-mediated transport of WPBs along microtubules. MTOC-localized WPBs are devoid of P-selectin, the adhesion receptor for leukocytes, but store cargo superfluous to an inflammatory response, for example angpt2. Thus, while histamine stimulates release of P-selectin from WPBs at the cell surface, thereby supporting an inflammatory reaction, histamine limits an excessive vascular injury response by anchoring WPBs containing cargo extraneous to inflammation to the MTOC. Dissociation of the perinuclear localized WPBs requires a continuous mobilization of intracellular Ca$^{2+}$, evoking binding of Ca$^{2+}$ to the EF-hand of Rab46. In this way, Rab46 integrates transport and Ca$^{2+}$ signals to regulate trafficking of WPBs, which is relevant to the physiological signal and thus limits the extensive emergency response evoked by vascular injury. We propose that understanding the molecular mechanisms underlying Rab46 function will provide targets for therapeutic intervention in cardiovascular disease.

## Results

### Rab46 is a novel Rab GTPase localized to WPBs

To determine the role of Rab46 in ECs, we first investigated the subcellular localization of endogenous Rab46 in human umbilical vein endothelial cells (HUVECs) and human cardiac microvascular endothelial cells (HCMECs) using a previously validated antibody raised against Rab46 (Wilson et al., 2015). Immunofluorescent staining of Rab46 revealed a cytoplasmic granular pattern in both HUVECs (Fig. 1 a, green) and HCMECs (Fig. 1 b). The characteristic distribution and the presence of cigar-shaped vesicles led us to speculate that Rab46 was localized to WPBs, the endothelial-specific storage vesicle for the procoagulant vWF. Costaining with the WPB marker vWF (Fig. 1, a and b, red) revealed colocalization of Rab46 and vWF. Further analysis by Airyscan high-resolution imaging (Fig. 1 c) revealed that individual WPBs contain vWF juxtaposed to Rab46,

suggesting that Rab46 is external to vWF. In contrast, secretory granules containing tissue plasminogen activator, lamp-1–positive secretory lysosomes, or Rab11-positive recycling vesicles (Fig. 1 d) displayed no immunoreactivity to Rab46. In addition, overexpressed Rab46 localized to WPBs (Fig. 1 e). These data suggest that Rab46 is primarily targeted to WPBs.

To gain insight into the role of Rab46 in WPB function, we first investigated if Rab46 localized to all, or a subpopulation of, WPBs. Quantification of images immunostained for vWF and Rab46 revealed that not all WPBs contain Rab46; only 49% (±13%) of WPBs were positive for both vWF and Rab46 (Fig. 2 a). This suggests that Rab46 regulates a subset of WPBs. Next, we questioned if, in a manner similar to other Rab GTPases (Nightingale et al., 2009), Rab46 was necessary for WPB biogenesis. vWF is the determinant for WPB generation; therefore, we investigated vWF expression and WPB numbers in response to Rab46 depletion. Depletion of Rab46 by specifically targeted siRNA (example blot and mean data in Fig. S1) had no effect on the number of cells that expressed WPBs (Fig. 2 b, 88 ± 3.5%) or the expression of vWF mRNA (Fig. 2 c), indicating that Rab46 does not impair WPB biogenesis. However, siRNA-mediated depletion of Rab46 significantly increased the abundance of vWF protein in the cells (densitometry 1.6 ± 0.49: Fig. 2 d) with a concomitant increase in the number of WPBs per cell (control, 95 ± 6, vs. siRNA Rab46, 128 ± 8; Fig. 2 e). Taken together, these data suggested that Rab46 plays a role in the trafficking of a subpopulation of WPBs.

### Histamine induces a WPB perinuclear clustering that is dependent on Rab46

To address the role of Rab46 in agonist-dependent WPB trafficking, we investigated the effects of Rab46 depletion on WPB distribution in HUVECs stimulated with histamine (Figs. 3 and S1 c). Surprisingly, while vehicle-treated cells displayed an even distribution of Rab46 and WPBs throughout the cytoplasm (Fig. 3, a [images] and b [mean data]; and Fig. S1 c), acute histamine stimulation (10 min) induced significant perinuclear trafficking of both vWF and Rab46 (Fig. 3 a, bottom left). The depletion of Rab46 with specifically targeted siRNAs had no effect on the distribution of WPBs in basal conditions (Fig. 3 a, top right). However, histamine failed to induce perinuclear clustering of vWF in cells depleted of Rab46 (Fig. 3 a, bottom right), suggesting that histamine-evoked clustering of a subpopulation of WPBs to the perinucleus depends on Rab46 (Fig. 3, a and b; and Fig. S1 c). Histamine exerts its function by binding to four different G protein–coupled receptors (H$_1$–H$_4$; Panula et al., 2015). Using RT-PCR analysis, we identified that the major receptor expressed in HUVECs was H$_1$, with significantly lower levels of H$_4$ (Fig. 3 c; the cycle threshold values for H$_2$ and H$_3$ were too high for their presence to be confirmed). To characterize the receptor subtype responsible for histamine-induced perinuclear trafficking of WPBs, selective agonists to H$_1$ (2-pyridylethylamine dihydrochloride [2-PY]) or H$_4$ (4-methyl-histamine dihydrochloride [4-Met]) were used. 2-PY but not 4-Met induced perinuclear clusters of Rab46 and WPBs at the MTOC (Fig. 3 d), suggesting that histamine-evoked WPB trafficking is mediated by the H$_1$ receptor. This perinuclear

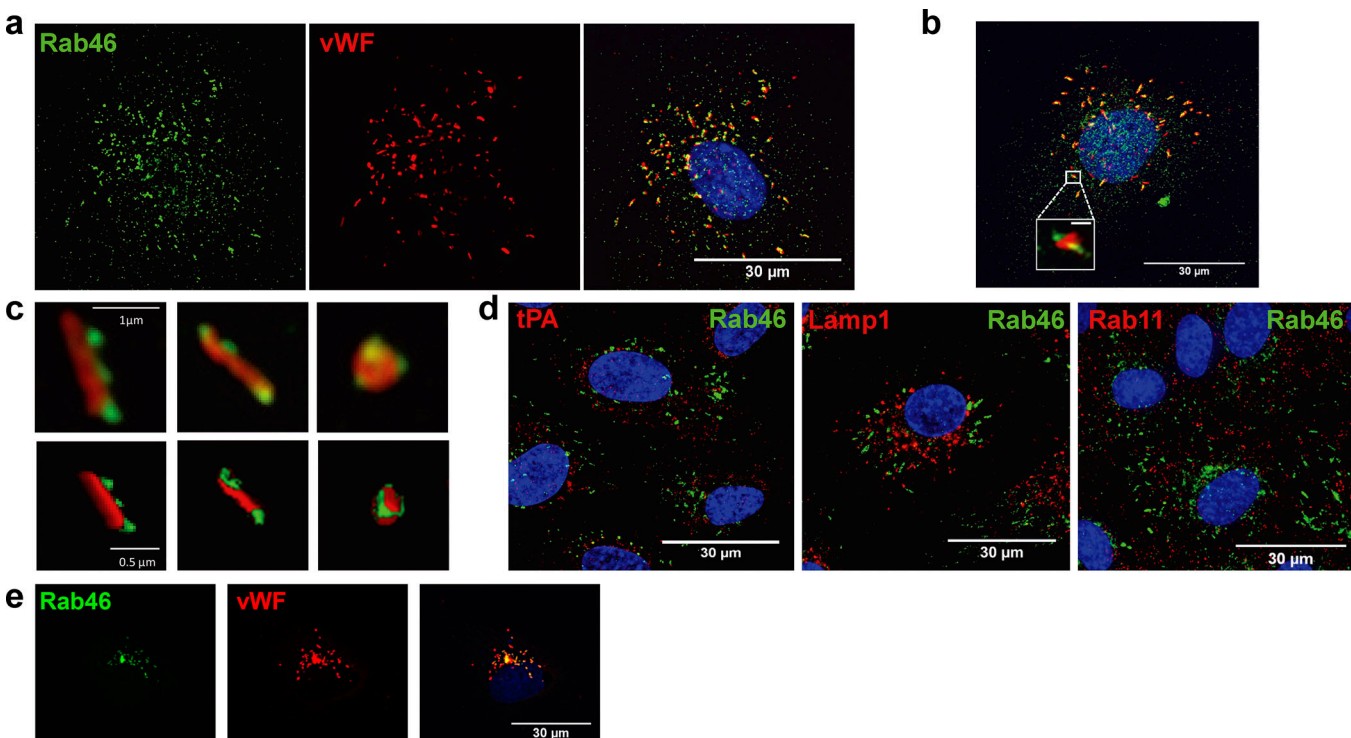

**Figure 1.  Rab46 localizes to WPBs. (a and b)** Immunofluorescent images showing subcellular localization of endogenous Rab46 (green) and vWF (red) in HUVECs (a) and HMCECs (b). Boxed area (b) shows enlargement of WPBs showing cigar-shaped organelle localized to Rab46 vesicles. Scale bar in boxed area = 1 µm. **(c)** High-resolution Airyscan imaging and 3D reconstruction (bottom) showing single WPBs (vWF, red) where Rab46 (green) is juxtaposed to vWF. **(d)** Confocal microscopy images showing subcellular localization of Rab46 with tissue plasminogen activator (tPA) vesicles (left), Lamp1 as lysosomal marker (middle), and Rab11-positive recycling endosomes (right). Merged images are used to show colocalization of Rab46 to WPBs and other vesicles. **(e)** HUVECs transfected with WT Rab46 (green, anti-Rab46) demonstrate Rab46 localized to WPBs (red, anti-vWF). Maximum-intensity projections from DeltaVision or confocal microscopy z stack are shown. Number of independent biological repeats/technical repeats = 3/6. Scale bar = 30 µm.

trafficking event was also specific to histamine and not thrombin (Fig. S2 a) and was time dependent (Fig. S2 b).

### Nucleotide binding is necessary for Rab46-dependent trafficking

Rab domains contain highly conserved nucleotide binding sites that dictate GTPase membrane localization and function (Dumas et al., 1999; Pereira-Leal and Seabra, 2000). Srikanth et al. (2016) have previously demonstrated that WT Rab46 has the intrinsic ability to hydrolyze GTP, while nucleotide-binding or GTPase mutants have less GTPase activity, suggesting that the GTPase domain is functionally active. We therefore predicted that nucleotide binding to the GTPase domains of Rab46 would be necessary for histamine-induced retrograde transport of WPBs. Consistent with endogenous Rab46 localization, heterologously expressed WT Rab46 localized to vWF-positive WPBs (Figs. 1 e and 4 a). Unlike endogenous Rab46, some of these Rab46-positive WPBs localized to a perinuclear region, but because many Rab proteins have slow intrinsic GTP hydrolysis activity, this overexpression could drive a GTP-bound active state. In support of Rab46 motility to the perinucleus being dependent on GTP binding, a Q604L constitutively active Rab46 mutant was localized to the perinuclear region (Figs. 4 a and S3 b). In agreement with GTP-bound Rab46 being necessary for WPB motility, vWF was colocalized with Q604L Rab46 at a

perinuclear region even in the absence of stimulation (Fig. S3 b). In contrast, both the nucleotide-free mutant (Fig. 4 a, N658I) and the inactive GDP-bound mutant (Fig. 4 a, T559N) were localized to the cytoplasm. Furthermore, histamine failed to induce redistribution of Rab46 or perinuclear clustering of WPBs in cells overexpressing the N658I inactive Rab46 mutant (Fig. 4 b). The data suggest that GTP binding is necessary for the perinuclear localization of Rab46 and WPBs.

### Histamine induces microtubule- and dynein-dependent WPB trafficking to the MTOC

We next sought to identify the WPB-localized perinuclear compartment evoked by histamine. Using Airyscan imaging, we established that histamine-evoked clusters of WPBs (Fig. 5 a, vWF) and Rab46 (Fig. 5 b) were localized to pericentrin, a marker corresponding to the MTOC. This localization was distinct from the ER or the Golgi (Fig. S3 a). In addition, the constitutively active Q604L Rab46 mutant localized to pericentrin and vWF in the absence of histamine stimulation (Fig. S3 b). These data demonstrate that histamine evoked Rab46-dependent trafficking of WPBs toward the MTOC. Since retrograde movement toward the centrosome is dependent on the integrity of the microtubules and the microtubule motor protein dynein, we analyzed histamine-induced WPB trafficking in the presence and absence of nocodazole or ciliobrevin-D,

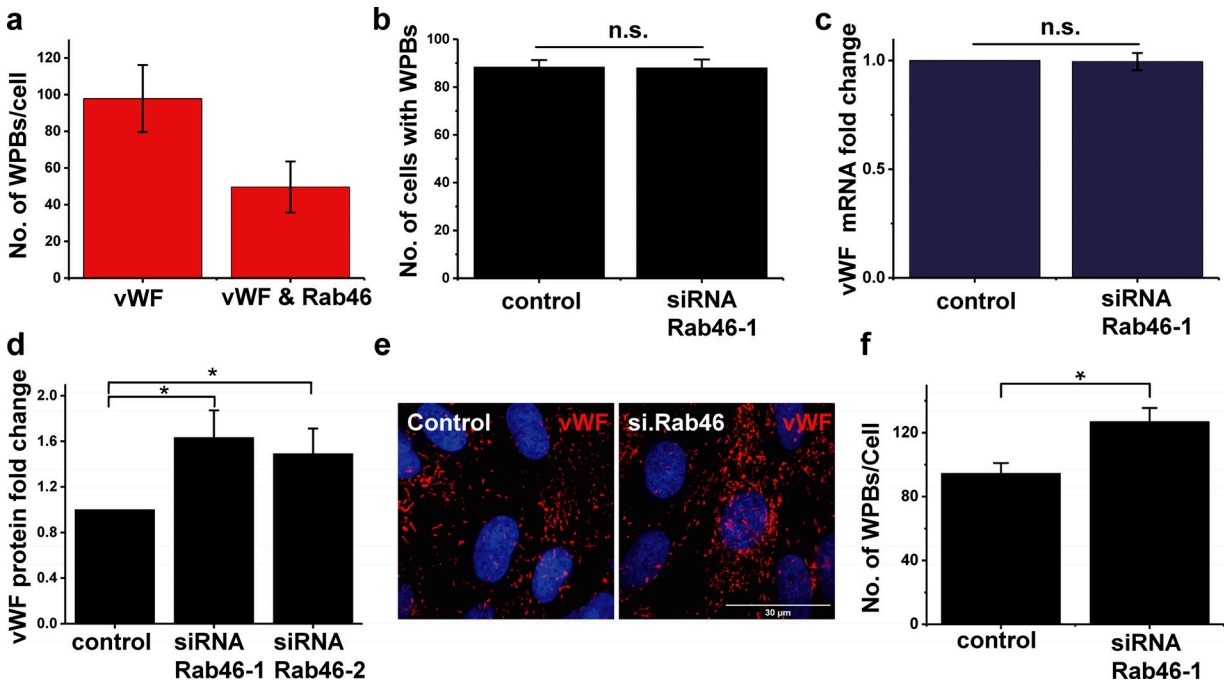

Figure 2. **Rab46 localizes to a subpopulation of WPBs and increases intracellular vWF content after depletion. (a)** The number of vWF-positive WPBs per cell were quantified from images and compared with the number of WPBs that were also positive for Rab46 staining (see Materials and methods). Quantification of the number of WPBs per cell associated with Rab46 (49 ± 13 WPBs of 98 ± 18) per cell (number of independent biological repeats/technical repeats = 3/5). **(b)** To quantify the percentage of cells that contain WPBs, we counted the number of vWF-positive vesicles observed in images. We quantified the number of cells positive for vWF in HUVECs when cells were depleted of Rab46 by targeted siRNA (siRNA Rab46-1) or a control siRNA (number of independent biological repeats/technical repeats = 3/18). **(c)** Quantitative PCR change in cycle threshold (ΔCT) analysis of HUVECs transfected with Rab46 siRNA-1 demonstrates no difference in the expression of vWF mRNA compared with cells transfected with control siRNA ($n = 3$). **(d)** Densitometry analysis from a Western blot of vWF band intensity in HUVECs transfected with two different siRNAs specific for Rab46 is shown as fold-change relative to housekeeping genes (siRNA Rab46-1, 1.63 ± 0.23; siRNA Rab46-2, 1.49 ± 0.22; $n = 6$). **(e)** Example images used to quantify cells stained for vWF as a marker for WPBs in the presence or absence of Rab46 (siRNA-1). **(f)** The number of WPBs per cell quantified as WPB counting described in Materials and methods, where cells were depleted of Rab46 versus siRNA control (control, 95 ± 6; siRNA Rab46-1, 128 ± 8). Number of independent biological repeats/technical repeats = 3/18. Graphs show mean ± SEM. n.s., not significant. *, $P < 0.05$ by Student's $t$ test.

respectively. Microtubule depolymerization by nocodazole or inhibition of the dynein complex by ciliobrevin-D abolished the histamine-induced perinuclear clustering of WPBs (Fig. 5, c and d).

To further explore the mechanisms underlying Rab46-dependent trafficking, we performed protein affinity purification followed by mass spectrometry analysis to identify interacting proteins. This analysis suggested dynein heavy chain (DHC) as a candidate effector protein. Immunoprecipitation and Western blot analysis verified the interaction between GFP-tagged constitutively active (Q604L) Rab46 with DHC (Fig. 6 a). Accordingly, immunoprecipitation experiments using anti-Rab46 (Fig. 6 b) and anti-DHC (Fig. 6 c), in both the presence and absence of histamine stimulation, confirmed the interaction between endogenous Rab46 and DHC. Taken together, these data suggest that Rab46 interacts with dynein and that GTP binding is necessary for trafficking along microtubules to the MTOC.

**Histamine-evoked perinuclear-localized WPBs are devoid of P-selectin**

To understand the physiological relevance of histamine-evoked perinuclear trafficking, we first considered that, as WPBs store both prothrombotic and proinflammatory cargo (Rondaij et al., 2006), there must be populations of WPBs storing mutually exclusive cargo coupled to agonist-evoked signaling pathways. Histamine is involved in inflammatory responses (Falus and Merétey, 1992), and one of the pivotal events required for the recruitment of neutrophils to sites of inflammation is the histamine-dependent release of P-selectin at the EC surface (Burns et al., 1999). Therefore, we explored the role of Rab46 in histamine-dependent release of P-selectin from WPBs. In control conditions, imaging analysis revealed that P-selectin localized to WPBs that were distinct from Rab46-positive WPBs (Fig. 7, a and b; and Fig. 8 c). While histamine stimulation evoked trafficking of Rab46-positive WPBs to the MTOC (Fig. 7 c, white arrow), P-selectin–carrying WPBs trafficked to the cell surface (Fig. 7 c, thick white arrow). Quantification of P-selectin released at the cell surface in HUVECs transfected with control or Rab46 siRNA demonstrated that histamine-evoked surface exposure of P-selectin was independent of Rab46 (Fig. 7 d). Furthermore, P-selectin did not localize with constitutively active Rab46 (Q604L) at the MTOC (Fig. 7 e).

Previously, P-selectin and the proangiogenic tie-2 ligand angpt2 have been shown to reside in mutually exclusive WPBs (Fiedler et al., 2004); therefore, we questioned whether the WPBs clustered at the perinucleus by histamine contained

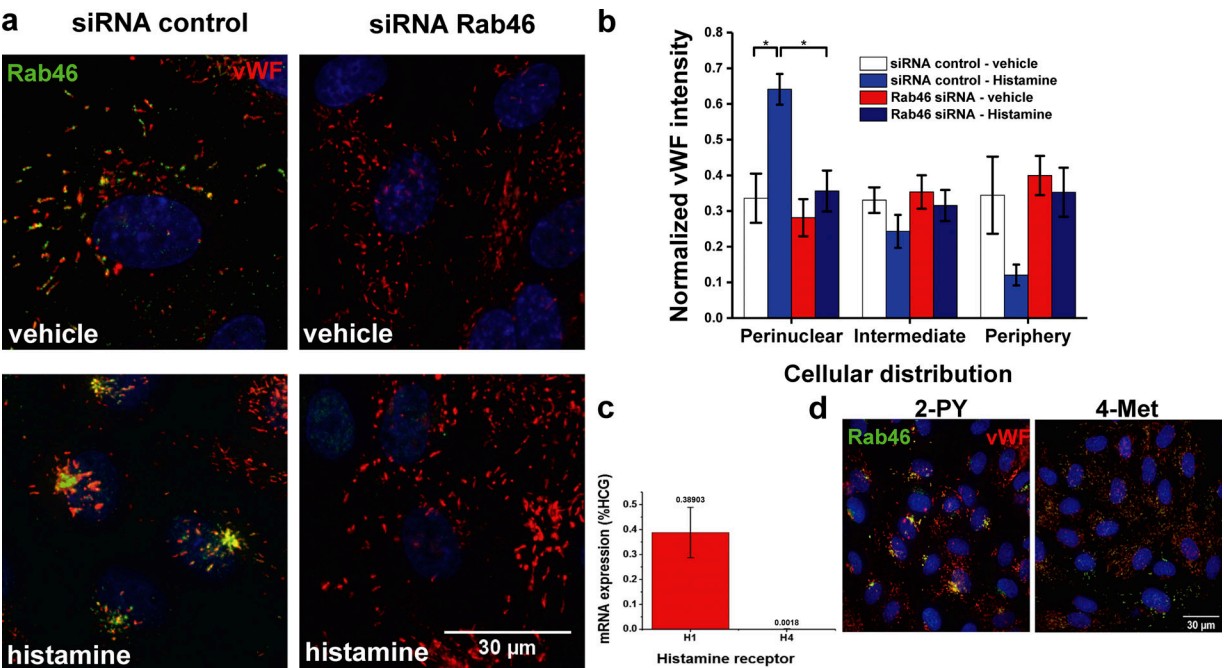

**Figure 3.** **Histamine-induced Rab46-dependent perinuclear clustering of WPBs. (a)** Representative images of HUVECs transfected with Rab46 siRNA-1 or control siRNA in control conditions (vehicle) or after 10-min histamine treatment (30 µM). Immunostaining of Rab46 (green) and vWF (red) shows perinuclear cluster of Rab46 and vWF in siRNA control cells upon histamine stimulation (bottom left). Perinuclear trafficking of WPBs is completely abolished in Rab46-depleted cells (bottom right). Scale bar = 30 µm. **(b)** Quantitative analysis of vWF cellular distribution in siRNA control cells and Rab46-depleted cells upon histamine stimulation. Results were grouped into three areas: perinuclear, intermediate, and periphery. The plot shows vWF signal intensity of each particle in the respective area where the mean (± SEM) was noted as percentage of the total signal intensity. Number of independent biological repeats/technical repeats = 3/36. *, P < 0.05 by two-way ANOVA. **(c and d)** Graph of quantitative PCR change in cycle threshold (ΔCT) analysis (± SEM) of histamine receptor mRNA expression compared with housekeeping control genes (HCG; n = 3; c) and representative images of cells immunostained for Rab46 (green) and vWF (red) and stimulated with 100 µM of the $H_1R$ agonist 2-PY or 100 µM of the $H_4R$ agonist 4-Met (d). Scale bar = 30 µm.

angpt2. First, we confirmed that P-selectin and angpt2 reside primarily in discrete WPBs in HUVECs (Fig. 8, a and b). Moreover, in basal and stimulated conditions, Rab46 localized to WPBs that were devoid of P-selectin (control, 8.32 ± 0.38%; histamine, 7.38 ±1.87%) but contained angpt2 (control, 27.58 ±1.48%; histamine, 32.27 ±1.52%; Fig. 8, c and d). Stimulation of HUVECs with histamine promoted the Rab46-dependent MTOC localization of WPBs carrying angpt2 (Fig. 8, d [arrow] and e [mean data]) but not P-selectin (Fig. 8 e, mean data; and Fig. 7 c). In addition, in the absence of stimulation, angpt2 is confined to the MTOC only when colocalized with constitutively active Rab46 (Fig. 8 f, Q604L). Rationalizing that full-blown exocytosis of angpt2 would be superfluous to an on-demand histamine response and that Rab46-dependent retention of angpt2-containing WPBs at the MTOC could play a role in constraining secretion, we investigated the effect of acute histamine stimulation on angpt2 secretion. Acute stimulation with a low dose (0.3 µM) of histamine that does not elicit WPB clustering at the MTOC (Fig. S4) evoked release of angpt2 from HUVECs (Fig. 8 g, blue column). However, stimulating cells with a higher (but physiological) dose (30 µM) of histamine failed to augment secretion (Fig. 8 g, red column), suggesting that histamine-evoked clustering of angpt2-positive WPBs at the MTOC prevents further secretion. We questioned if inhibition of Rab46 (thereby preventing perinuclear WPB clustering) would enhance angpt2 secretion. Surprisingly, angpt2 levels were not

increased in supernatants from histamine-stimulated cells where Rab46 was depleted (Fig. 8 g, hatched columns). The comparable effect of Rab46 depletion on angpt2 secretion in PMA-stimulated cells led us to question the role of Rab46 on angpt2 protein. Depletion of Rab46 in HUVECs had no effect on angpt2 mRNA expression (Fig. 8 h) and only a small effect on angpt2 protein levels (Fig. 8 i). High-resolution imaging demonstrated that in cells where Rab46 was depleted (Fig. 8 j, right), angpt2 was no longer localized to WPBs. Taken together, these data support the concept that Rab46 couples inflammatory stimuli to WPB trafficking to limit release, and Rab46 may be necessary for angpt2 recruitment.

## Histamine-evoked trafficking to the MTOC is cAMP and $Ca^{2+}$ independent

Rab46 is an unusual Rab GTPase because, in addition to its nucleotide-binding activities (Wilson et al., 2015), it has the putative capability to respond to $Ca^{2+}$ signals via its EF-hand ($Ca^{2+}$-binding domain). We examined whether $Ca^{2+}$ regulates histamine-stimulated WPB trafficking to the MTOC. First, we showed that histamine evoked a dose-dependent increase in intracellular $Ca^{2+}$ with a maximum response evoked by 30 µM histamine (Fig. 9 a). To determine which pools of $Ca^{2+}$ are mobilized by histamine, we stimulated cells with histamine in the absence of extracellular $Ca^{2+}$ (to eliminate $Ca^{2+}$ entry from the extracellular space), and then $Ca^{2+}$ was added back as a measure

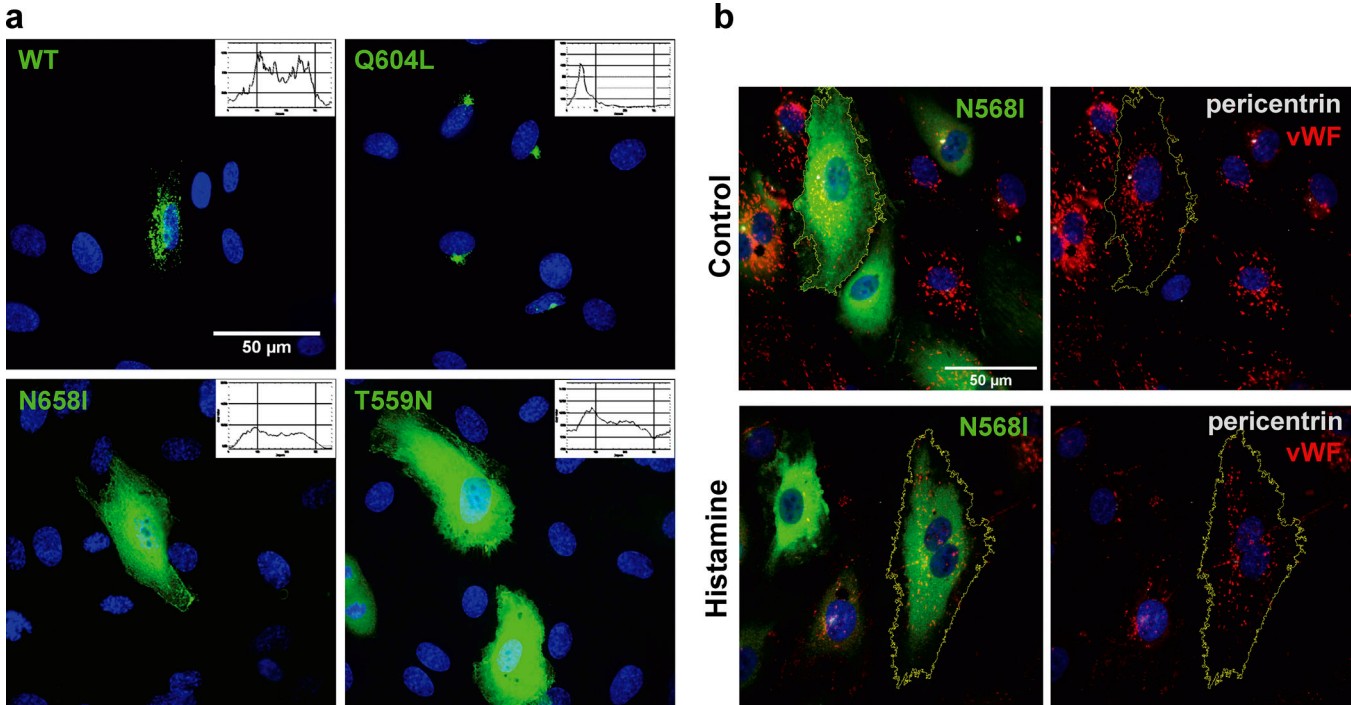

Figure 4. **Subcellular localization of WT and Rab46 nucleotide binding mutants. (a)** Representative image of HUVECs expressing WT Rab46 and a representative intensity plot with a random distribution (top left). Representative image of constitutively active form of Rab46 (Q604L) with its intensity plot where the peak indicates clustering (top right). Representative images of the nucleotide-free mutant (N658I; bottom left) and the inactive GDP-bound form of Rab46 (T559N; bottom right) expressed in HUVECs and their representative intensity plots showing homogeneous green distribution indicating a cytosolic localization. Plots were generated using the Oval profile method in ImageJ. Number of independent biological repeats/technical repeats = 3/28. **(b)** Histamine does not induce perinuclear clustering of cells overexpressing the inactive N658I Rab46 mutant. Cells treated with vehicle or 30 µM histamine (10 min) and immunostained for Rab46 (green), vWF (red), and pericentrin (white) as a marker of the MTOC. Example cells outlined in yellow. Number of independent biological repeats/technical repeats = 3/15. Scale bar = 50 µm.

of store-operated $Ca^{2+}$ entry through the Orai-1 channel (Fig. 9 b). Under $Ca^{2+}$-free conditions, 30 µM histamine induced a transient peak response after 10 s, demonstrating $Ca^{2+}$ release from intracellular stores. Pretreatment of HUVECs with the membrane-permeable fast $Ca^{2+}$ chelator 1,2-bis(2-amino-phenoxy)ethane-$N,N,N',N'$-tetraacetic acid tetrakis(acetoxy-methyl ester) (AM-BAPTA) significantly inhibited the histamine-induced release from intracellular stores and from the extracellular space (Fig. 9, c and d).

The necessity for an increase in intracellular $Ca^{2+}$ in WPB secretion is well documented (Zupancic et al., 2002; Erent et al., 2007). To determine if histamine-evoked WPB perinuclear trafficking was dependent on mobilization of intracellular $Ca^{2+}$, we observed Rab46 localization in HUVECs that were preloaded with AM-BAPTA. Chelation of intracellular $Ca^{2+}$ had no effect on histamine-induced trafficking of WPBs to the MTOC, suggesting this is a $Ca^{2+}$-independent event (Fig. 9, e and f). In addition, mutating the $Ca^{2+}$-binding sites in the EF-hand of Rab46 (Rab46$^{EFmut}$) enhanced the MTOC localization in nonstimulated conditions (Fig. 9 g). These data support the concept that $Ca^{2+}$ is not necessary for the histamine-evoked retrograde trafficking of Rab46 and WPBs to the MTOC.

Several agonists evoke WPB perinuclear trafficking via stimulation of intracellular cAMP (Kaufmann and Vischer, 2003; Xiong et al., 2009; Brandherm et al., 2013). Although

histamine-dependent Rab46 perinuclear trafficking is via the $H_1$ receptor (a Gq-coupled receptor; Fig. 3 c), we sought to determine if cAMP was necessary for histamine-induced trafficking. Preincubation of cells with the protein kinase A (PKA) inhibitor H-89 before histamine stimulation did not inhibit histamine-evoked $Ca^{2+}$ entry (Fig. S5 a) or perinuclear trafficking of Rab46 (Fig. S5 b), suggesting this pathway does not depend on cAMP. We then sought to determine if Rab46 played a role in cAMP-dependent trafficking. Acute stimulation of cells with epinephrine (and the phosphodiesterase inhibitor, 3-isobutyl-1-methylxanthine [IBMX]) did not provoke robust clustering in HUVECs (Fig. S5 c). However, clustering was still observed in cells depleted of Rab46 by targeted siRNA (Fig. S5 c, arrows).

### Intracellular $Ca^{2+}$ evokes Rab46-dependent WPB dispersal

If histamine-evoked WPB trafficking to the MTOC is $Ca^{2+}$ independent, we questioned the function of the $Ca^{2+}$-sensing domains (EF-hands) in Rab46. We know that a shorter isoform of the Rab46 gene (CRACR2A-S), which lacks the Rab domain but has functional EF-hands, has important roles in T cells. Srikanth et al. (2010) have demonstrated that $Ca^{2+}$ binds to the EF-hands of CRACR2A-S, and this evokes dissociation of proteins clustered at the plasma membrane. In support of $Ca^{2+}$ binding to the EF-hand being necessary for dissociation of Rab46 clusters at the MTOC, we show that, in dividing cells, Rab46$^{EFmut}$ was anchored

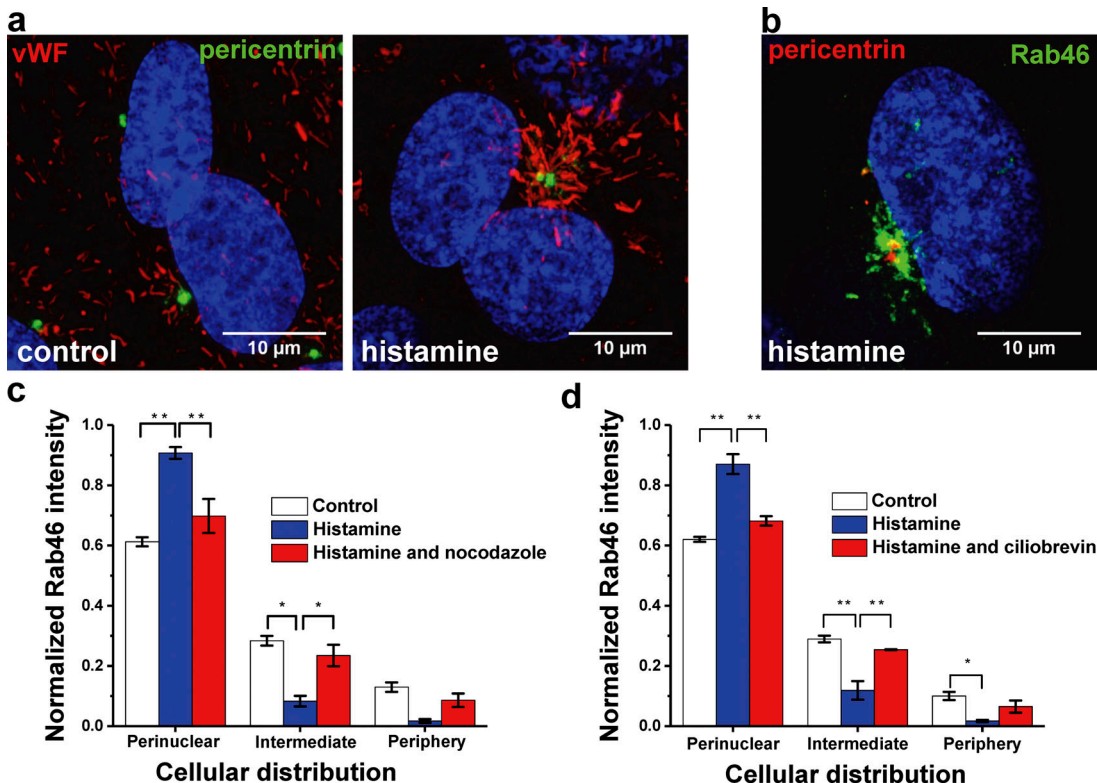

Figure 5. **The integrity of microtubules and dynein is necessary for Rab46-dependent trafficking of WPBs to the MTOC. (a)** Airyscan images of HUVECs stained with vWF (red) and pericentrin (green). Control cells on the left and 30 µM histamine–treated cells on the right showing reorientation of WPBs toward the MTOC. **(b)** High-resolution Airyscan image of cells treated with 30 µM histamine (10 min) and then immunostained with Rab46 (green) and pericentrin (red) showing Rab46 clustering at the MTOC. Maximum-intensity projections from confocal microscopy z stack are shown. **(c and d)** Mean data showing the cellular distribution of Rab46 upon histamine stimulation in cells pretreated with nocodazole (c) or ciliobrevin (d) and respective control cells. The plots quantify Rab46 signal intensity in the respective area where the mean (± SEM) is noted as a percentage of the total signal intensity. Number of independent biological repeats/technical repeats = 3/18; *, $P < 0.05$; **, $P < 0.01$ by one-way ANOVA.

to mitotic spindles, indicating that $Ca^{2+}$ may be necessary for detachment from the microtubules and release from the MTOC (Fig. 10 a, arrows). We also observed Rab46-positive mitotic spindles in cells expressing N-terminal GFP-tagged WT Rab46, suggesting steric hindrance of the EF-hand by GFP. We therefore predicted that mobilization of intracellular $Ca^{2+}$ may be necessary for dispersal of histamine-induced WPB perinuclear clusters.

EF-hand domains will bind to any locally available free $Ca^{2+}$ ion (Gifford et al., 2007). To investigate the effect of global free $Ca^{2+}$ ions on the distribution of WPBs, we first stimulated HUVECs with histamine to induce WPB trafficking to the MTOC, then we forced global intracellular $Ca^{2+}$ release with the sarco/endoplasmic reticulum $Ca^{2+}$-ATPase pump inhibitor thapsigargin and observed the effects on the dissociation of Rab46 clusters. Thapsigargin treatment alone had no effect on the trafficking of WPBs but triggered dispersal of the histamine-evoked perinuclear clusters of endogenous Rab46 (Fig. 10, b and c, mean data). To investigate if this was due to $Ca^{2+}$ binding to the EF-hand of Rab46, we performed the same experiment in HUVECs overexpressing WT Rab46 or EF-hand mutated Rab46 (Fig. 10, d and e, mean data). Thapsigargin induced the dispersal of WT Rab46 but not the EF-hand mutant, suggesting that it is $Ca^{2+}$ binding to the EF-hand of Rab46 that is necessary for dispersal of Rab46 from the perinuclear clusters.

## Discussion

In this study, we demonstrate that acute histamine stimulation evokes Rab46-dependent WPB trafficking. We propose a model whereby histamine evokes the exocytosis of P-selectin–carrying WPBs at the plasma membrane. Concomitantly, histamine induces activation of Rab46, thus promoting Rab46 interaction with dynein and trafficking of a subpopulation of WPBs that are devoid of P-selectin but contain other cargo, such as angpt2, to the MTOC. WPBs cluster at the MTOC, where the EF-hand of Rab46 binds to $Ca^{2+}$ released locally, thus inducing dissociation of Rab46 from the microtubules and dispersal of WPBs.

Histamine, an amine involved in transient immune and inflammatory responses, evokes leukocyte attraction without the necessity for EC migration or vasoconstriction (Pober and Sessa, 2007; Sun et al., 2012). It is vital, therefore, that ECs couple physiological stimuli to the selective release of WPB cargo to achieve a functionally appropriate response. Recent studies have described mechanisms underlying the differential release of cargo from WPBs docked at the cell surface. A lingering kiss during weak histamine stimulation excludes the release of cargo >40 kD (Babich et al., 2008), while agonist-dependent recruitment of an actomyosin ring controls the force necessary for expulsion of the vWF moiety without affecting release of other cargo (Nightingale et al., 2018). To date, there has been no

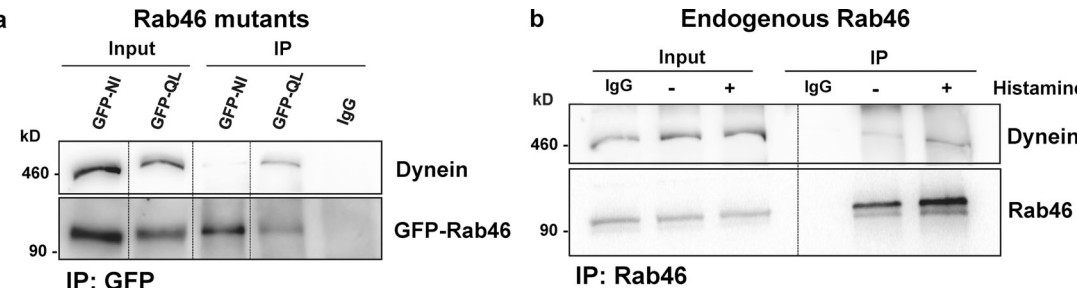

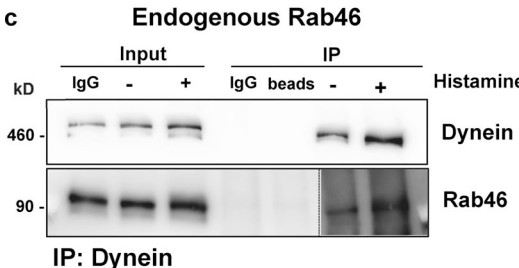

**Figure 6.** **Rab46 interacts with the DHC. (a)** GFP-tag active and inactive forms of Rab46 (Q604L and N658I, respectively) were overexpressed in HUVECs, and immunoprecipitation (IP) was performed using an anti-GFP antibody. Western blot analysis shows that the active form of Rab46 (Q604L) binds to dynein. **(b)** Immunoprecipitation of endogenous Rab46 in HUVECs performed using an anti-Rab46 antibody; coprecipitation of DHC was assessed by immunoblotting for DHC. **(c)** Reverse immunoprecipitation of endogenous Rab46 performed using anti-DHC antibody; coprecipitation was assessed by immunoblotting for Rab46. Rab46 is shown at an increased exposure and overlaid on the blot. Blots are representative of three independent experiments.

description of differential storage and release of cargo in a contextual responsive process or if the cellular machinery exists to support such regulation.

For the first time, we demonstrate that WPBs storing the cargo proteins P-selectin and angpt2 are differentially trafficked in response to acute histamine stimulation. Leukocyte attraction to the endothelium occurs in minutes (Sun et al., 2012). Thereby, we describe a strategy where an on-demand immune response evoked by histamine in the absence of vascular injury would allow P-selectin to be released at the cell surface while redirecting WPBs carrying functionally irrelevant cargo, for example angpt2, to the MTOC (thereby allowing leukocyte recruitment without inducing cell migration).

Next, we show that trafficking of these distinct WPB populations is contextual, since perinuclear clustering of P-selectin–negative WPBs is specific to histamine but not thrombin. Fiedler et al. (2004) demonstrated that thrombin but not histamine evoked angpt2 secretion from HUVECs, indicating that transient anchoring of WPBs at the MTOC rescues angpt2 from immediate release. Here we demonstrate that histamine-evoked perinuclear clustering as a mechanism to restrict release of angpt2 and possibly other cargo is not necessary for an acute inflammatory response. However, trafficking of cargo-restricted WPBs to the MTOC may also regulate polarized secretion (Lopes da Silva and Cutler, 2016).

Histamine-evoked trafficking of WPBs is dependent on Rab46, a novel Rab GTPase that we discovered in ECs (Wilson et al., 2015). Rab46 is localized to a subpopulation of WPBs that are P-selectin negative. We show that constitutively active Rab46 mimics histamine stimulation and evokes trafficking of these discrete WPBs to the MTOC. We suggest that histamine acts via

the $H_1$ receptor on the EC surface, evoking a signaling pathway that activates Rab46. We are currently exploring the mechanisms underlying Rab46 recruitment to WPBs, but this precise recruitment together with GTP binding presents an explanation of how histamine can stimulate release of P-selectin at the cell surface (Rab46 independent) while inducing WPBs carrying angpt2 to the MTOC (via Rab46 activation). The effect of Rab46 depletion on vWF protein levels suggests that Rab46 may also play a role in exocytosis, which needs further exploration.

Histamine activation of Rab46 induces dynein-dependent trafficking of WPBs along microtubules toward the MTOC. In a similar manner, Wang et al. (2019) proposed that, in T cells, Rab46 is necessary for dynein-dependent transport to the MTOC; furthermore, they suggested that Rab46 directly recruits dynein in a $Ca^{2+}$-dependent manner. In ECs, we demonstrate that dynein-mediated WPB transport is $Ca^{2+}$ independent; moreover, $Ca^{2+}$ binding to Rab46 is necessary for detachment from the microtubules. A mutation of Rab46 that is unable to bind to $Ca^{2+}$ (Rab46[EFmut]) traffics to the MTOC, where it becomes anchored. Unlike endogenous or WT Rab46, this is not dispersed by increases in global $Ca^{2+}$. Additionally, in some dividing cells, heterologous expression of Rab46[EFmut], but not other Rab46 mutants, provokes generation of mitotic spindles, indicating that Rab46[EFmut] remains attached to dynein, thereby delocalizing dynein during spindle formation (Quintyne et al., 2005). This difference in $Ca^{2+}$ dependence might reflect differences in the physiology between the two cell types. Upon T cell activation, the MTOC and Rab46 traffic to the immunological synapse, and this is necessary for signaling between the T cell receptor and the JNK pathway (Srikanth et al., 2016). In addition, while ECs only express the Rab46 (cracr2a-a: CRACR2A-L)

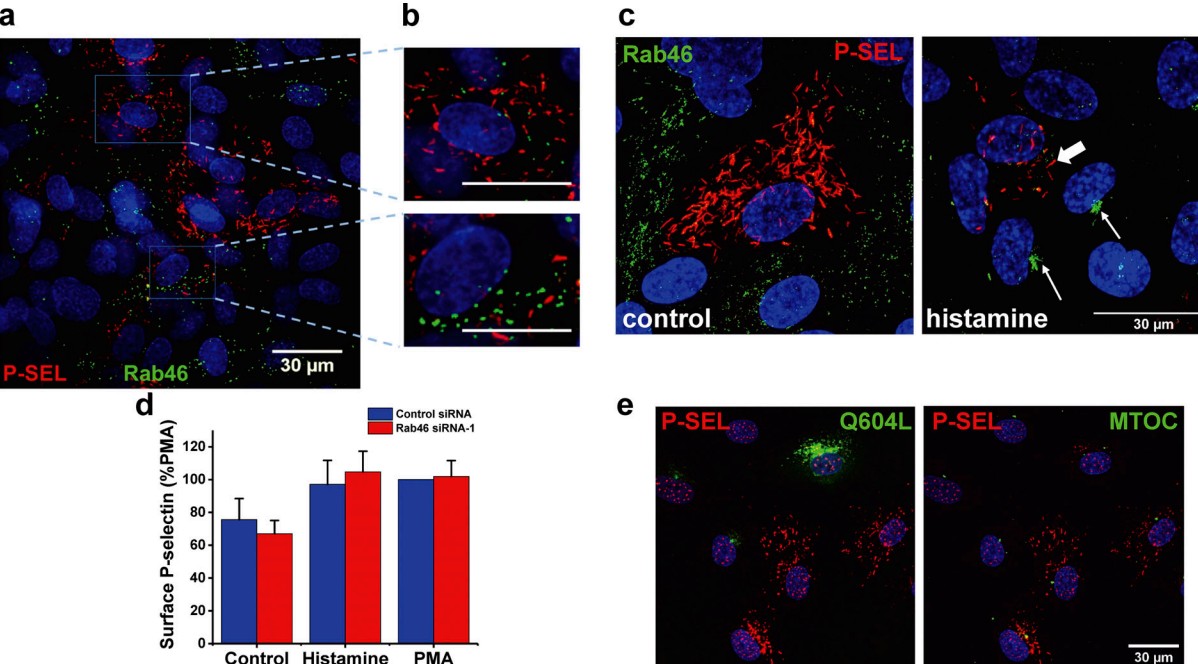

Figure 7.   **Histamine-evoked cell surface expression of P-selectin. (a)** Images of HUVECs immunostained for P-selectin (P-SEL; red) and Rab46 (green). The blue boxes represent the areas magnified in b. **(b)** P-selectin and Rab46 in distinct WPBs in the same cell. Scale bars = 30 µm. **(c)** Airyscan images of HUVECs immunostained for P-selectin (P-SEL; red) and Rab46 (green) in response to control or 5-min treatment with 30 µM histamine. Thin white arrows, clusters of Rab46-positive WPBs; thick arrow, cell surface P-selectin. Scale bar = 30 µm. **(d)** ELISA measurements of cell surface P-selectin expression in HUVECs transfected with either control or Rab46 siRNA-1. HUVECs were stimulated with vehicle, 30 µM histamine, or 1 µM PMA. Plot indicates mean ± SEM percentage of maximal control siRNA response (PMA). Number of independent biological repeats/technical repeats = 3/9. **(e)** Constitutively active Rab46 does not localize P-selectin at the MTOC. Cells overexpressing the constitutively active Q604L Rab46 mutant (green), P-selectin (red; left image), and pericentrin as a marker for the MTOC (red; right image). $n$ = 3.

isoform of the EFCAB4B gene, T cells also express a short non-Rab isoform (*cracr2a-c*: CRACR2A-S) that lacks the Rab domain but can interact with Rab46 via the coil-coiled region. Both Rab46 and *cracr2a-c* are necessary for regulating store-operated $Ca^{2+}$ entry (SOCE) in T cells, but we found no function of Rab46 in SOCE in ECs (Wilson et al., 2015). Experimentally, differences in Rab46 localization may exist due to the overexpression of Rab46 fused to a GFP tag (N-termini). We have noted that GFP-tagged WT Rab46 is similar to Rab46$^{EFmut}$, indicating that GFP hinders the EF-hand and thereby excludes the use of this tag for observing Rab46 dynamics.

We have demonstrated that Rab46 labels a P-selectin–negative population of WPBs, and its activity is coupled to stimuli. However, Rab46 also contains an EF-hand that can bind $Ca^{2+}$ (Srikanth et al., 2010); therefore, we questioned the role of $Ca^{2+}$ in histamine-evoked WPB trafficking. We show that $Ca^{2+}$ binding to the EF-hand of Rab46 is necessary for the dispersal of the MTOC-bound Rab46. A key question remains regarding the function of WPB/Rab46 dispersal. We speculate that this could be a means of replenishing the cells with WPBs and/or Rab46 ready for further cellular insult or a method of redirecting WPBs toward the apical surface to allow polarized secretion. We also question the need to integrate $Ca^{2+}$ and GTPase activity on one protein. Both thrombin and histamine elicit $Ca^{2+}$ signals in ECs. Specificity of cellular responses to $Ca^{2+}$ signals is achieved by spatial and temporal regulation of the release of intracellular $Ca^{2+}$. The small but significant response to global $Ca^{2+}$ release and

the precise localization of histamine-stimulated Rab46 indicate the involvement of highly localized $Ca^{2+}$ signaling events. Thereby, we suggest that GTPase activity both provides spatial control and dictates the confirmation of Rab46 so that the EF-hand is revealed in the correct locale. In this way, Rab46 integrates both GTPase transport signals and $Ca^{2+}$ signals to deliver an appropriate response.

It is evident that a deficiency of vWF causes a number of diseases; however, increased plasma levels of other cargo stored in WPBs are reported in cardiovascular and neoplastic diseases, and in the aging population they contribute to an increased risk of thrombosis (Lerman et al., 1991; Vischer, 2006; Alfonso and Angiolillo, 2013; Korhonen et al., 2016; Schuldt et al., 2018). The fundamental mechanisms by which physiologically diverse agonists coordinate intracellular signals to discriminate between cargo-restricted populations of WPBs has not previously been described. Here, we present Rab46 as a stimulus-coupled regulator of WPB trafficking that could be a novel target for the development of cardiovascular disease therapeutics.

## Materials and methods

### Cell culture

Pooled HUVECs (Lonza or Promocell) were grown in endothelial basal cell medium 2 supplemented with Endothelial Growth Medium-2 Singlequot supplements (Lonza), maintained at 37°C in a humidified atmosphere of 5% $CO_2$, and used between

Figure 8. **Histamine induces differential trafficking of P-selectin–negative WPBs that contain angpt2. (a)** P-selectin (red) and angpt2 (green) reside in mutually exclusive WPBs. **(b)** Representative high-resolution Airyscan image shows P-selectin (red) and angpt2 (green) distribution in WPBs. **(c)** Quantification of percentage Rab46 colocalization to P-selectin– and angpt2-positive WPBs in control and 30 µM histamine–treated HUVECs. The plot indicates the mean ± SEM; number of independent biological repeats/technical repeats = 3/18–30. **(d)** Images of HUVECs immunostained for endogenous angpt2 (red) and Rab46 (green) in response to control or 5-min 30 µM histamine treatment. Arrow depicts coclusters of Rab46 and angpt2. Scale bar = 30 µm, applies to both images. **(e)** P-selectin, angpt2, and Rab46 cellular distribution in response to 5-min treatment with 30 µM histamine. Plot represents mean ± SEM normalized intensity in the respective area; number of independent biological repeats/technical repeats = 5/28–40. **(f)** Angpt2 (red) localizes to pericentrin (green; white arrow) in the absence of stimulation when expressed in cells overexpressing constitutively active Q604L Rab46 (green). Angpt2 does not cluster in cells not overexpressing Q604L (white thick arrow). **(g)** ELISA-based analysis of angpt2 secretion shown as a percentage of the PMA response in cells treated with siRNA control or Rab46 siRNA-1. Quantification was performed on supernatants collected from HUVECs treated with vehicle, 0.3 µM histamine, 30 µM histamine, or 1 µM PMA. Number of independent biological repeats/technical repeats = 3/6. **(h)** ΔPCR change in cycle threshold (ΔCT) analysis of HUVECs transfected with Rab46 siRNA-1 demonstrates no difference in the expression of angpt2 mRNA compared with control siRNA ($n = 3$). **(i)** Western blot densitometry of angpt2 band intensity in HUVECs transfected with siRNA specific for Rab46 is shown as fold-change relative to housekeeping genes (siRNA Rab46-1, 0.7 ± 0.14; $n = 3$). **(j)** Representative images depicting loss of angpt2 localization to WPBs when cells have been depleted of Rab46 (siRNA Rab46-1). Scale bar = 30 µm. n.s., not significant. ***, $P < 0.001$ by one- or two-way ANOVA as appropriate.

passages 1 and 5. HCMECs (PromoCell) were grown in Endothelial Cell Growth Medium MV2, maintained at 37°C in a humidified atmosphere of 5% $CO_2$, and used between passages 1 and 7.

## siRNA transfection

Control siRNA, EFCAB4B (Rab46), and angpt2 siRNA Silencer Select were obtained from Ambion (see Table S4 for sequences). P-Selectin and control siRNA were obtained from Dharmacon (see Table S4 for sequences). HUVECs at 80–90% confluence were used for transfection, which was performed using a 1:3 ratio of 50 nmol/liter siRNAs with Lipofectamine 2000 reagent (Invitrogen) diluted in OptiMEM (Gibco) as per the manufacturers' instructions. Cells were incubated with the transfection solution for 6 h before medium change. Knockdown was assessed via Western blotting. All experiments were performed at 48 and 72 h after transfection.

## cDNA transfection

HUVECs were plated into Ibidi µ-slide 8-well plates ($7 \times 10^4$ cells/ml, 300 µl per well) and transfected after 24 h using Lipofectamine 2000 in Opti-MEM. A 3:1 ratio between Lipofectamine and cDNA (100 ng) was used.

## Rab46 mutagenesis

Various mutants of Rab46 were generated by PCR amplification (pHusion DNA polymerase) and site-directed mutagenesis using primers described in Table S1.

## RT-PCR

RNA was extracted using TRI Reagent (Sigma-Aldrich). Reverse transcription (Applied Biosystems) was followed by real-time PCR using SYBR green using a LightCycler (Roche). Primers are provided in Table S5.

## Western blotting

Cells were seeded at $4 \times 10^5$ cells/well and harvested with 100 µl lysis buffer (10 mM Tris, pH 7.5, 150 MmM NaCl, 0.5 mM EDTA, 0.5% NP-40, and protease and phosphatase inhibitor cocktails [Sigma-Aldrich]). Proteins were separated by SDS-PAGE (7.5%

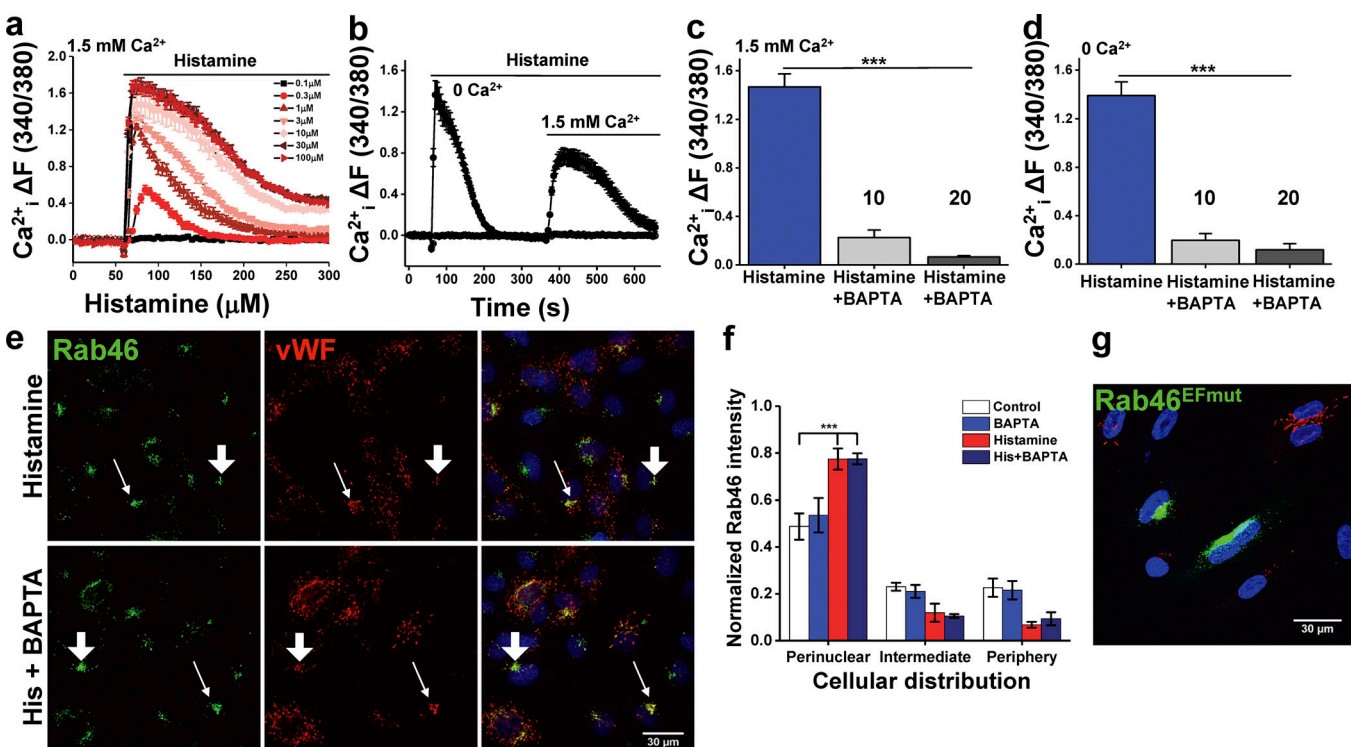

Figure 9. **Acute histamine treatment–induced perinuclear localization of WPBs is independent of Ca²⁺. (a)** Dose response of histamine-induced Ca²⁺ mobilization. Representative traces of change (Δ) in intracellular Ca²⁺ evoked by histamine in HUVECs loaded with the Ca²⁺ indicator Fura-2-AM. Number of independent biological repeats/technical repeats = 5/30. **(b)** Histamine evokes a biphasic SOCE response in HUVECs. In the absence of extracellular Ca²⁺, the first phase reflects the release of Ca²⁺ from intracellular stores, while the addition of 1.5 mM extracellular Ca²⁺ to the recording solution evokes a second phase that reflects the influx of extracellular Ca²⁺. Number of independent biological repeats/technical repeats = 5/36. **(c)** Mean data of peak intracellular Ca²⁺ evoked by histamine in HUVECs pretreated with vehicle or 10 or 20 μM AM-BAPTA in the presence of 1.5 mM extracellular Ca²⁺. Number of independent biological repeats/technical repeats = 3/9. **(d)** Mean data of peak intracellular Ca²⁺ evoked by histamine in HUVECs pretreated with vehicle or 10 or 20 μM AM-BAPTA in the absence of extracellular Ca²⁺. Number of independent biological repeats/technical repeats = 3/9. **(e)** Representative images of HUVECs immunostained for vWF (red) and Rab46 (green) in response to 30 μM histamine (10 min) ± pretreatment with 10 μM AM-BAPTA. Scale bar = 30 μm. **(f)** Mean data of Rab46 cellular distribution in response to 30 μM histamine treatment for 5 min in HUVECs pretreated with either vehicle or 10 μM AM-BAPTA. The plot represents mean ± SEM normalized intensity in the respective area; number of independent biological repeats/technical repeats = 3/18–30. **(g)** Representative image of HUVECs expressing a mutant of Rab46 mutant that cannot bind Ca²⁺ (Rab46^EFmut, green) costained with vWF (red). Scale bar = 30 μm. ***, P < 0.001 by one-way ANOVA.

or 4–20% TGX Gels) and transferred to PVDF membrane using Mini Trans-Blot Cell (Bio-Rad). Membranes were incubated for 1 h in blocking solution consisting of 5% (wt/vol) milk diluted in TBS-T (145 mM NaCl, 20 mM Tris-base, pH 7.4, and 0.5% Tween-20) and labeled with primary antibody overnight at 4°C for vWF (Dako), Rab46 (CRACR2A: Proteintech), Vinculin (Sigma-Aldrich), angpt2 (R&D Systems), P-selectin (Santa Cruz), and GAPDH (GeneTex). Immunoblots were visualized using HRP–conjugated donkey anti-mouse, anti-rabbit, or anti-goat secondary antibodies (Jackson ImmunoResearch Labs; see Table S3) and SuperSignal Femto (Pierce). Protein band densities were quantified with ImageJ (National Institutes of Health).

### Intracellular Ca²⁺ measurement

Intracellular Ca²⁺ was measured using a multimodal microplate reader, Flexstation II³⁸⁴ (Molecular Devices). The change (Δ) in intracellular Ca²⁺ concentration above baseline was shown by the ratio of fura-2-acetoxymethyl ester (Fura-2-AM) fluorescence emission for 340- and 380-nm excitation. The fluorescence ratio of ΔF 340/380 was used as an analysis measure.

HUVECs were seeded (2.5 × 10⁴ cells/well) onto clear-bottomed Nunc 96-well plates (Thermo Fisher Scientific) and allowed to reach confluence for 24 h. Cells were incubated with 2 μM Fura-2-AM loading solution containing 0.01% pluronic acid in standard bath solution consisting of 130 mM NaCl, 5 mM KCl, 1.2 mM MgCl₂, 1.5 mM CaCl₂, 8 mM D-glucose, and 10 mM Hepes, pH 7.4, for 60 min at 37°C. Cells were washed in standard bath solution at room temperature for 30 min before experiments.

### Immunoprecipitation

HUVECs plated in 10-cm² Petri dishes were washed with PBS, cross-linked with 1% PFA, and then harvested with 250 μl lysis buffer (NP-40). The lysate was quantified, and 0.5 mg of total lysate was incubated with 1 μg of antibody or control IgG for 4 h at 4°C with rotation. This mixture was then added to 30 μl of preequilibrated Protein G sepharose beads (GE Healthcare Life Science) and incubated overnight under continuous agitation. Beads were washed thoroughly before elution of the bound proteins in 20 μl of 4× sample buffer solution (200 mM Tris, pH 6.8, 8% SDS, 40% glycerol, 8% mercaptoethanol, and 0.1%

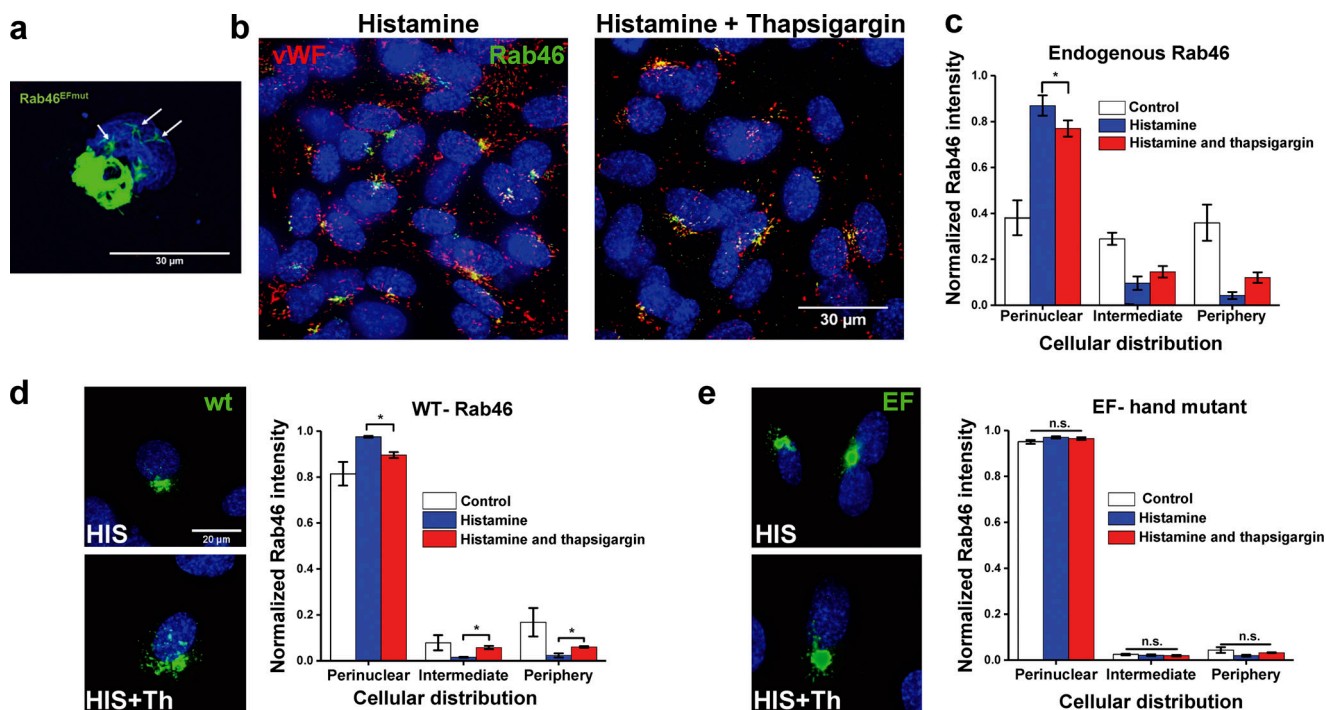

Figure 10. **Intracellular Ca²⁺ evokes Rab46-dependent WPB dispersal from the MTOC. (a)** Example of EF-hand and multimitotic spindles. Immunostaining of HUVECs overexpressing a mutation of Rab46 that cannot bind Ca²⁺ (Rab46$^{EFmut}$, green), where Rab46 is localized to multiple spindle-like compartments (arrows). Scale bar = 30 µm. **(b)** Representative images of HUVECs treated with 30 µM histamine for 10 min alone or followed by thapsigargin (1 µM) and stained for endogenous Rab46 (green) and vWF (red). **(c)** Quantitative analysis of endogenous Rab46 dispersal shown in b, showing cellular distribution of Rab46. **(d and e)** Representative images and quantitative analysis of histamine-stimulated cells overexpressing WT (d) and EF-hand Rab46 mutant (e) ± thapsigargin (Th) showing Rab46 intensity distribution into three different cellular areas after stimulation with histamine or histamine followed by thapsigargin. Scale bar = 20 µm, applies to all images in d and e. The plots show Rab46 signal intensity in the respective areas, where the mean (± SEM) was noted as percentage of the total signal intensity. Number of independent biological repeats/technical repeats = 3/30. *, P < 0.05; n.s., not significant by one-way ANOVA.

bromophenol blue) and boiled at 95°C for 5 min. The elution fraction was loaded onto an SDS-PAGE gel, and the amounts of Rab46 and DHC were detected by Western blot.

### P-selectin cell surface ELISA

HUVECs were seeded in 96-well plates at a cell density of 5 × 10⁴/well and grown to confluence for 72 h with daily medium change. HUVECs were serum starved for 1 h before secretagogue treatment in serum-free M199 plus 10 mM Hepes. All reagents were brought to room temperature before the experiment. After respective treatments were terminated, cells were washed with 1× PBS and incubated with 4% PFA for 5 min, followed by a blocking step with BSA. Incubation with P-selectin primary antibody (Santa Cruz; sc-19672) and HRP-conjugated secondary antibody (Jackson ImmunoResearch Labs; 115-035-003) solution was performed at room temperature. 3,3′,5,5′-Tetramethylbenzidine substrate (Sigma-Aldrich; T0440) was added for 5 min with gentle agitation until the characteristic color change was obtained. The reaction was terminated with an equal amount of 0.5 M H₂SO₄ and read at 450 nm on the Flexstation II³⁸⁴.

### Angpt2 ELISA

HUVECs were grown for 72 h until formation of confluent monolayers. Before the experiments, cells were starved in serum-free M199 plus 10 mM Hepes for 1 h. The medium was fully removed, and cells were treated with histamine, DMSO, and PMA, all diluted in fresh M199 plus 10 mM Hepes. Only M199 plus 10 mM Hepes was applied on control cells. Cells supernatant was collected at room temperature and stored at −20°C for later analysis. The angpt2 concentration in cell supernatant was measured using Human Angpt2 Quantikine ELISA Kit (R&D Systems). ELISA was performed according to manufacturer instructions.

### Immunocytochemistry

Cells were seeded at 8 × 10⁴ cells/ml into Ibidi µ-slide eight-well plates. HUVECs were grown for 72 h. Cells were starved in serum-free M199 plus 10 mM Hepes medium for 1 h before treatments. Histamine was used at 30 µM unless otherwise stated, epinephrine hydrochloride (Sigma-Aldrich; E4642) was used at 100 µM in solution with 100 µM IBMX (Sigma-Aldrich; I5879), and thrombin (Sigma-Aldrich; T1063) was used at 2.5 U/ml. Cells were pretreated with 1 µM nocodazole (Sigma-Aldrich; SML1665), 40 µM ciliobrevin (Sigma-Aldrich; 25401), 10 µM AM-BAPTA (Tocris; 2787), and 10 µM H89 (Tocris; 2910/1) in related experiments before histamine stimulation. Cells were fixed with 4% PFA for 10 min, washed with PBS, and permeabilized with 0.1% Triton X-100 solution. Cells were incubated with primary antibody (in PBS; see Table S2 for primary

antibodies) for 1 h followed by fluorescently labeled appropriate species secondary antibodies for 30 min (see Table S3 for secondary antibodies). Cells were briefly incubated in Hoechst before being mounted with Ibidi mounting medium.

### DeltaVision wide-field deconvolution microscopy
Cells were visualized on an Olympus IX-70 inverted microscope using 20×/0.75 and 40×/1.35 oil objectives supported by a DeltaVision deconvolution system (Applied Precision) with SoftWorx image acquisition and analysis software. 10 focal planes at 0.2 μm per z-stack were taken using a Roper CoolSNAP HQ charge-coupled device camera. Iterative deconvolution (5×) was performed on z stacks using the proprietary algorithm. The filter sets used were DAPI, FITC, and TRITC. All imaging was performed at room temperature.

### High-resolution Airyscan microscopy
High-resolution microscopy was performed using an inverted confocal laser-scanning microscope, Zeiss LS880 with Airyscan system. Images were captured using a 63×/1.4 oil objective and 405-nm diode, Argon/2 (458, 477, 488, and 514 nm), HeNe 543 nm, and HeNe 633 nm lasers. All the images were acquired and processed with Zen software. All imaging was performed at room temperature.

### Image analysis
Maximum-intensity projections and 3D surface rendering images were performed using DeltaVision Softworx or Zen accordingly and analyzed in ImageJ Fiji. In ImageJ, the channels were split, then vWF and Rab46 channels were subjected to background subtraction depending on the noise level. The noise was estimated using a region of interest in the background for measurement, and the mean was subtracted from the entire image. Pixels whose intensity values were similar to the background were replaced with the mean background intensity value.

### WPBs counting
For counting of WPBs, segmentation of WPBs was applied. A local threshold algorithm (Bernsen method) with 15-pixel radius was applied, and a binary image was created. The number of objectives was calculated according to their size in every image and normalized to the number of nuclei per image. The number of vWF-positive cells and the number of nuclei were determined using a cell counter plugin.

### Colocalization analysis
WPBs marked with vWF antibody and endogenous Rab46 fluorescent structures were quantified using ICY software (Spot detector plugin). WPBs and Rab46 vesicles were delineated by ~14 pixel and 4–7 pixel spot size, respectively. Detection of Rab46 (green spot) only over WPBs structures (red spot) was performed by selecting the red channel as driver channel and the region-of-interest module to detect the green spots.

### Analysis of the cellular distribution of Rab46 mutants
The pixel intensity of Rab46 mutants in the perinuclear area was measured using the Oval profile plugin in ImageJ (https:// imagej.nih.gov/ij/plugins/oval-profile.html). First, an oval was drawn around the nucleus and enlarged by 20%, and then the plugin, with the option radial sum, was executed to create a pixel intensity profile plot for each cell.

### WPB and Rab46 distribution
Cellular distribution and particle intensity were determined using Fiji. A customized macro (Data S1) was designed to automate the analysis. Briefly, a 16-bit image was loaded in Fiji. The channels were split, and a binary mask was created using a Default threshold on the DAPI channel. Noise was reduced using a median filter, and if required, adjacent sites were split using the watershed algorithm. A distance map was generated. Green and red channels were duplicated to sample the original pixel intensities of WPBs and Rab46 structures. Each channel was segmented using a threshold algorithm (Max Entropy), and the distance and intensity of each particle from the nucleus were measured. The macro automatically exports results tables with distance and intensity values for each particle, binary images of the distance map, and .tiff images of each channel per analyzed image. 5–10 wide-field images (40×) were analyzed per experimental group. Distance (Min) and integrated intensity (IntDen) values were imported to OriginPro as x and y values, respectively. Distance values ranged from 0 px (nucleus) to 255 px (periphery). This list of numerical values was divided into three bins: perinuclear ($x < 2$ μm), intermediate ($2$ μm $< x < 5$ μm), and periphery ($x > 5$ μm), where $x$ is the distance from the nucleus to the cell periphery. The integrated intensity was normalized by the total fluorescence intensity of the image. The mean values were calculated for each area and averaged among all analyzed images. The mean values averaged among all the biological repeats are presented as bar plots with mean ± SEM, where the y axis denotes normalized (integrated) intensity and the x axis denotes the distance from the nucleus, indicated by the average distribution of the analyzed signal based on the three bins in the analyzed cell population.

### P-selectin/angpt-2 colocalization
P-selectin and angpt2 colocalization with Rab46 was quantified using the spot colocalization ImageJ plugin ComDet (https:// imagej.net/Spots_colocalization_(ComDet)).

### Statistics
All average data are represented by mean ± SEM. Paired $t$ test was performed as appropriate when comparison among two data groups was sought. For comparison among three or more mean values, one- or two-way ANOVA was performed to determine whether significant differences exist among groups, coupled with Bonferroni post hoc test. Statistical significance was considered to exist at $P < 0.05$ (*, $P < 0.05$; **, $P < 0.01$; and ***, $P < 0.001$). Where comparisons lack an asterisk, they were not significantly different and/or marked as not significant. OriginPro 2017 software was used for data analysis and presentation.

### Online supplemental material
Data S1 is an ImageJ macro used for analyzing the cellular distribution of Rab46 and vWF. Table S1 is a list of primers used for

Rab46 mutagenesis. Table S2 lists antibodies used for immunofluorescence and Western blotting. Table S3 lists secondary antibodies, and Table S4 lists the siRNA sequences. Table S5 lists the primers used for RT-PCR. Fig. S1 shows an example Western blot and quantification of the effect of two separate Rab46-targeted siRNAs on Rab46 protein expression and the effect of Rab46 siRNA-2 on histamine-evoked vWF cellular distribution. Fig. S2 shows mean data analysis of histamine and thrombin-evoked vWF cellular distribution over time. Fig. S3 shows immunofluorescent images of histamine-evoked Rab46 cellular localization compared with ER and Golgi (a) and constitutively active Rab46 localized to vWF and pericentrin (b). Fig. S4 shows example Western blots of P-selectin and angpt2 detection, with and without specifically targeted siRNAs, and immunofluorescent images of Rab46 and vWF localization in response to 0.3 µm histamine. Fig. S5 shows calcium traces and mean data of Rab46 cellular distribution in the presence of a PKA inhibitor to support that the histamine response is not dependent on cAMP. In addition, immunofluorescent images show that inhibition of Rab46 does not affect epinephrine-induced clustering.

## Acknowledgments

We thank Leeds Bioimaging Facility staff for their support.

This study was supported by funding from the Medical Research Council with a New Investigator Research Grant MR/N000285/1 awarded to L. McKeown; British Heart Foundation nonclinical PhD studentship FS/17/43/33003 to K.T. Miteva; and British Heart Foundation nonclinical PhD studentship FS/14/22/30734 to H.J. Gaunt. D. Sobradillo was supported by a short-term fellowship from the Junta para la Ampliación de Estudios predoctoral program from Consejo Superior de Investigaciones Científicas.

The authors declare no competing financial interests.

Author contributions: L. McKeown conceived the study, designed experiments, performed experiments, and analyzed the results. L. Pedicini, L.A. Wilson, and K.T. Miteva performed imaging experiments and analyzed results. L. Pedicini, H.J. Gaunt, and K.T. Miteva performed Ca$^{2+}$ imaging, ELISAs, and Western blotting experiments. K. Marszalek, F. Bartoli, and R.G. Slip performed quantitative PCR. S. Deivasigamani, D. Sobradillo, L. McKeown, and L. Pedicini undertook the molecular biology. L. McKeown wrote the paper. I. Jayasinghe designed the image analysis. D.J. Beech provided intellectual input. All authors reviewed and approved the final version of the manuscript.

Submitted: 24 October 2018

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
