## [Reviewer comments · The Journal of Cell Biology]

Rab46 integrates Ca²⁺ and histamine signaling to regulate selective cargo release from Weibel-Palade bodies

Katarina Miteva, Lucia Pedicini, Lesley Wilson, Izzy Jayasinghe, Raphael Slip, Katarzyna Marszalek, Hannah Gaunt, Fiona Bartoli, Shruthi Deivasigamani, Diego Sobradillo, David Beech, and Lynn Mckeown

Corresponding Author(s): Lynn Mckeown, University of Leeds

Review Timeline:

Submission Date:	2018-10-24
Editorial Decision:	2018-11-23
Revision Received:	2019-03-24
Editorial Decision:	2019-04-11
Revision Received:	2019-04-17

Monitoring Editor: Pier Paolo Di Fiore

Scientific Editor: Melina Casadio

Transaction Report:

DOI: 10.1083/jcb.201810118

November 26, 2018

Re: JCB manuscript #201810118

Dr. Lynn Mckeown
University of Leeds
LICAMM
Clarendon Way
Leeds LS2 9JT
United Kingdom

Dear Dr. Mckeown,

Thank you for submitting your manuscript entitled "A Ca²⁺-regulated G protein (Rab46) couples physiological stimuli to selective release of cargo from Weibel-Palade bodies". Your manuscript has been assessed by expert reviewers, whose comments are appended below. Although the reviewers express potential interest in this work, significant concerns unfortunately preclude publication of the current version of the manuscript in JCB.

You will see that the reviewers find the results interesting and Reviewers #1 and #3 in particular are supportive of the work at the journal. However, they request more convincing evidence to strengthen your mechanistic model and the understanding of Rab46/CRACR2A-L function at the molecular level, and to establish the physiological role of Rab46/CRACR2A-L in Weibel-Palade body release. We have discussed all their comments and feel that these important points from experts in the field are valid and would need to be addressed for publication in JCB.

Rev#2 shared that, in their view, the literature has not established that CRACR2A-L/Rab46 functions as a Rab and requested experimental evidence to establish this. We encourage you to determine whether the paper from Srikanth et al (2016, *Sci Signaling*) and/or other works satisfactorily demonstrated that Rab46 is indeed endowed with the biochemical properties that one would expect of a Rab or whether more experimental evidence is needed to address this valid point. This point could be addressed in the text if the appropriate evidence has been published previously. On the other hand, we would strongly encourage you to tackle all the other points from these experts experimentally to the best of your ability. Addressing the reviewers' comments in full will help to develop the mechanistic aspects of the work, ensure the data is sound and rigorous (see the reviewers' requests for antibody specificity controls, knockdown efficiency controls, rescue experiments, etc.), and establish the physiological relevance. Please also address Rev#2's comments about manuscript presentation and citation of the literature with revisions to the text.

Please let us know if you are able to address the major issues outlined above and wish to submit a revised manuscript to JCB. Note that a substantial amount of additional experimental data likely would be needed to satisfactorily address the concerns of the reviewers. Our typical timeframe for revisions is three to four months; if submitted within this timeframe, novelty will not be reassessed. We would be open to resubmission at a later date; however, please note that priority and novelty would be reassessed.

If you choose to revise and resubmit your manuscript, please also attend to the following editorial

points. Please direct any editorial questions to the journal office.

GENERAL GUIDELINES:

Text limits: Character count is < 40,000, not including spaces. Count includes title page, abstract, introduction, results, discussion, acknowledgments, and figure legends. Count does not include materials and methods, references, tables, or supplemental legends.

Figures: Your manuscript may have up to 10 main text figures. To avoid delays in production, figures must be prepared according to the policies outlined in our Instructions to Authors, under Data Presentation, <http://jcb.rupress.org/site/misc/ifora.xhtml>. All figures in accepted manuscripts will be screened prior to publication.

Supplemental information: There are strict limits on the allowable amount of supplemental data. Your manuscript may have up to 5 supplemental figures. Up to 10 supplemental videos or flash animations are allowed. A summary of all supplemental material should appear at the end of the Materials and methods section.

If you choose to resubmit, please include a cover letter addressing the reviewers' comments point by point. Please also highlight all changes in the text of the manuscript.

Regardless of how you choose to proceed, we hope that the comments below will prove constructive as your work progresses. We would be happy to discuss them further once you've had a chance to consider the points raised. You can contact the journal office with any questions, cellbio@rockefeller.edu or call (212) 327-8588.

Thank you for thinking of JCB as an appropriate place to publish your work.

Sincerely,

Pier Paolo Di Fiore, MD, PhD
Editor, Journal of Cell Biology

Melina Casadio, PhD
Senior Scientific Editor, Journal of Cell Biology

Reviewer #1 (Comments to the Authors (Required)):

In 2015, Wilson et al. reported on a long isoform of CRACR2A that has 2 EF hands, and a "Rab" domain. In June of this year, Vale's lab described two proteins, Rab45 and the long form that these authors are now calling Rab46—they reported these proteins as adaptors for dynein and endocytosis. Srikanth et al. (2016) showed that Rab46 is important for T cell function and signaling.

Here, the authors show that Rab46 co-localizes with 49% of Weibel Palade bodies in endothelial

cells; histamine causes these to move the the perinuclear region via a Rab46 dependent process. Rab46 was not seen in Weibel Palade bodies containing P-selectin, and histamine caused the P-selectin granules to stay peripheral. Interestingly, the dispersal of perinuclear granules is the calcium sensitive feature.

The story is interesting-two granule classes differentially mobilized--and suitable for JCB after correction of the following minor errors. The model is far from the data and would be better if it focused on two classes of granules rather than a histamine stimulated GEF that is worth looking for, but premature for a model.

1. Please provide accession number for Rab45 and Rab46 sequences (upon first mention) so readers can find them in gene expression datasets
2. GTPase activity is not what is shown as being needed here (Page 2 and page 4 header and page 7 text); mutant data are consistent with GTP binding being needed (as correctly stated on page 4 body of paragraph). Please correct throughout.
3. Page 7 first paragraph starts with: Histamine evokes nucleotide activation of Rab46. This has not yet been shown!

Reviewer #2 (Comments to the Authors (Required)):

The manuscript by Miteva et al describes how a protein that they have named Rab46 regulates the trafficking and release of selective cargo from Weibel-Palade bodies (WPBs). There are some very interesting aspects to this manuscript, specifically the demonstration of a mechanism to explain how the pro-inflammatory mediator histamine stimulates release of P-selectin from WPBs whereas WPBs containing cargo extraneous to inflammation are instead anchored at the MTOC. The manuscript has a number of important issues that need to be addressed.

The introduction is poorly written. The authors refer to the protein under study, CRACR2A-L as Rab46. While there is some evidence that this protein, which is much larger than conventional Rabs may indeed have Rab-like functions this is not well described and the paper that best suggests Rab-like function is not cited. It is incumbent upon the authors to provide the experimental evidence that convinces the reader that indeed this protein is a Rab, even if atypical.

On pg. 2, the following line "Several studies have shown that calmodulin, when activated by Ca²⁺, activates Rab GEFs 10" is misleading. First, they say several studies and then reference one primary paper, and the paper cited does not show calmodulin interaction with GEF. There is no introduction of Weibel-Palade bodies. For a reviewer its frustrating to have to construct one's own introduction in order to fully evaluate the manuscript.

Specific comments:

In figure 1 the authors use an antibody against CRACR2A-L in immunofluorescence studies but provide no evidence that it is specific. In figure 3 they do provide evidence of antibody specificity but do not discuss as such. They should provide antibody validation in advance of the first use using loss of function controls. As is, only 1 siRNA is shown, the siRNAs are only moderately efficient, it is unclear which siRNA is used, only a single cell is shown.

The immunoblot in suppl. Fig. 1 is poor, is the inefficient knockdown a consequence of a partially

selective antibody or partially effective siRNAs. Moreover, in figure 3, the immunofluorescent signal is completely lost.

Due to the low quality of manuscript presentation it is difficult to fully understand the exact nature of the primary data for figure 2a and 2d. This data should be presented and better explained.

Figure 3 demonstrates that knockdown of CRACR2A-L blocks histamine-stimulated translocation of vWF (in the WPBs) to the perinuclear region. This is interesting but needs to be better controlled. Use both siRNAs, they have a different knockdown efficiency, do they give a different phenotype/strength of phenotype? A functional rescue with wild-type CRACR2A-L compared to mutations that presumably influence GTP status.

In figure 4a, it's not that Rab46 is not localized to WPBs containing P-selectin, it's that P-selectin and Rab46 are expressed in different cells. This makes little sense and is not consistent with the conclusions.

I don't understand the relationship of the short isoform to the G-domain-containing isoform in figure 8, why is it discussed? More importantly, the authors state that thapsigargin induces dispersal of histamine-clustered WPBs. Presumably this is based on the small decrease in the perinuclear concentration of endogenous Rab46, and yet there is no corresponding significant increase in the peripheral pool (Fig. 8b).

The model in figure 9 indicates a direct interaction between dynein (not labeled in the legend) and the Rab-like domain of CRACR2A-L. This is readily testable.

Reviewer #3 (Comments to the Authors (Required)):

The manuscript by Miteva and co-workers describes the regulation of trafficking of specific storage organelles designated Weibel-Palade bodies by the Ca²⁺-regulated GTP binding protein Rab46 in endothelial cells. Previously, Rab46 or CRACR2A has been shown to regulate Ca²⁺ entry in T cells. The current manuscript provides evidence for Rab46-dependent trafficking of Weibel-Palade bodies to the microtubule organizing centre (MTOC) in endothelial cells. Overall, the current manuscript introduces Rab46 as a novel player in the trafficking of storage-organelles and thereby describes a novel regulatory function for this unusual EF-hand containing GTP binding protein. In view of this newly reported role for Rab46 the current manuscript is of interest to the general readership of the Journal of Cell Biology.

Previously accumulation of Weibel-Palade bodies at the MTOC has been reported by Vischer and co-workers (Vischer et al. 2000) as well as Rondaij and co-workers (Rondaij et al, 2006). Interestingly, in these studies clustering of Weibel-Palade bodies was induced by the administration of agonists like epinephrine and forskolin which raise intracellular cAMP-levels. In the current study clustering of Weibel-Palade bodies at the MTOC was induced by histamine. Inclusion of stimulation-experiments employing epinephrine or forskolin is needed to establish whether Rab46 only promotes histamine-induced clustering.

The claim that Rab46 couples physiological stimuli to selective release of cargo from Weibel-Palade bodies is not sufficiently supported by the data. Currently, this is only based on the presence of P-selectin and angiopoietin-2 in distinct subset of Weibel-Palade bodies. As shown by van Agtmaal

and co-workers (van Agtmaal et al., 2012) this is probably based on a staining-artefact. In order to maintain this conclusion additional experimental approaches are needed. These must preferably include immuno-EM stainings for P-selectin and angiopoietin-2 as well as release studies showing that following histamine stimulation only P-selectin and not angiopoietin-2 is released. Control stimulation employing thrombin are needed to substantiate that under these conditions both P-selectin and angiopoietin-2 are released.

In the last part of the manuscript over-expression of Rab46 variants in endothelial cells is being employed. As can be seen from the images shown in Figure 6-8; the number of Weibel-Palade bodies is greatly reduced in transfected cells. This limits the conclusions that can be derived from these experiments. Preferably, viral systems providing a more homogenous expression should be used for these experiments.

Further more detailed comments are listed below.

The authors show that Rab46 is present on Weibel-Palade bodies. Based on their high-resolution imaging it appears that Rab46 is not evenly distributed on Weibel-Palade bodies. The images shown in Figure 1a reveals that Rab46 is also localized on another subcellular compartment. Did the authors attempt to define to which other subcellular compartment Rab46 is localized? In Figure 2 the authors show that Rab46 is only present on around 50% of Weibel-Palade bodies. This observation is based on co-localization studies; additional information is needed to show that only a subset of Weibel-Palade bodies contain Rab46; electron microscopy studies would probably be useful. Also this part of the study needs to be complemented with real-time analysis of Weibel-Palade body trafficking to demonstrate whether only Rab46-positive Weibel-Palade can traffick to the MTOC. It would also be needed to show how Weibel-Palade bodies acquire Rab46. Does only a subset of Weibel-Palade bodies acquire Rab46? How is this regulated? Which effectors mediate Rab46-dependent trafficking to the microtubule organizing center?

In figure 2 the authors report that siRNA-mediated down-regulation of Rab46 affects the VWF content in endothelial cells. They also report an increase in the number of Weibel-Palade bodies under these conditions. Did the authors investigate whether changes in the multimer pattern of VWF occur under these conditions? It is well-established that the propensity of VWF to form polymers is highly dependent on its ability to be stored in Weibel-Palade bodies.

In Figure 3 the authors show that siRNA-mediated down-regulation of Rab46 prevent histamine-induced clustering of Weibel-Palade bodies. In contrast, they report that thrombin does not induce clustering. This is surprising since thrombin has also been reported to increase intracellular Ca^{2+} levels thereby potentially activating Rab46 dependent clustering of Weibel-Palade bodies. As mentioned earlier cAMP-dependent clustering of Weibel-Palade bodies has been previously reported. It is crucial to address whether epinephrine or forskolin-induced clustering is also impaired in endothelial cells that lack Rab46.

The data reported in Figure 4 need to be improved. First of all conflicting data have been reported with respect to the co-localisation of P-selectin and angiopoietin-2 in Weibel-Palade bodies. An early study (Fiedler et al., 2004) showed that P-selectin and angiopoietin-2 are stored in a mutually exclusive manner in Weibel-Palade bodies. In a later study, clear co-localisation of angiopoietin-2 and P-selectin was observed (Van Agtmaal et al., 2012). First of all the authors need to show co-stainings of angiopoietin-2 and P-selectin. The analysis shown in Figure 4A reveals that under the specific conditions of this experiment only a very limited of cells express P-selectin. P-selectin expression can be upregulated by pre-incubating with IL-4 and this would provide a means to

obtain a more homogenous expression of P-selectin in endothelial cells. Also ultrastructural studies employing immune electron microscopy are needed to show that P-selectin and Rab46 do not co-localize on WPBs. In addition to histamine and PMA induced P-selectin expression also histamine and PMA-induced release of angiotensin-2 needs to be measured in cells treated with Rab46 siRNA.

In figure 4E quite some difference is observed between the P-selectin intensity of histamine and non-treated cells. The observed difference is not significant; larger numbers of cells will need to be analyzed in order to establish whether P-selectin containing Weibel-Palade bodies are not trafficking to the MTOC.

In general the current evidence for differential release of P-selectin and angiotensin-2 and its control by Rab46 is insufficiently supported by experimental data.

Data contained in Figure 6 are of interest. Results are obtained employing transient expression of Rab46 variants. Therefore only a limited number of cells are expressing the Rab46 variants. Using these methods only morphological analyses are feasible. The confocal images displayed in this figure need to be extended to also show whether Weibel-Palade bodies are present and whether they are localized to the MTOC or in the periphery of the cell. Stainings should preferably be performed for VWF, P-selectin and angiotensin-2; also the MTOC needs to be visualized in transfected cells.

Ideally viral delivery systems (adenovirus & lentivirus have been developed for endothelial cells) would allow for performing quantitative biochemical assays to monitor the effect of the Rab46 variants on Weibel-Palade body release and clustering.

In figure 7 AM-BAPTA is being used to interrogate whether intracellular Ca^{2+} is needed for clustering of Weibel-Palade bodies. Following incubation with AM-BAPTA Rab46 is present at the MTOC; it is not clear from the images shown in Figure 7e whether Weibel-Palade bodies also cluster at the MTOC in the presence of AM-BAPTA since the cells in which Rab46 localizes to the MTOC contain a very limited number of Weibel-Palade bodies. Also in Figure 7 cells that express Rab46EFmut display a very limited number of Weibel-Palade bodies. The lack of sufficient number of Weibel-Palade bodies in transfected cells does not allow for monitoring of the effect of the Rab46EF mutant on Weibel-Palade body clustering.

In Figure 8 primary data are not shown for cells expressing WT-Rab46 and EF-Hand mutant. The effects of thapsigargin treatment as shown in Figure 8c are very small. The data indicate that at best thapsigargin only very limitedly affects Rab46 localization in histamine-treated cells. In addition a marker for Weibel-Palade bodies needs to be added to these images in order to monitor the effect of thapsigargin, WT-Rab46 and EF-Hand mutant on the number and localization of Weibel-Palade bodies.

As mentioned previously data from two other research groups have shown that clustering of Weibel-Palade bodies at the MTOC occurs in response to agonists like epinephrine and forskolin. These agents raise intracellular cAMP levels. In view of these earlier data it would be useful to also explore whether Rab46 promotes cAMP dependent clustering of Weibel-Palade bodies.

We would like to thank all the reviewers for all their constructive comments which we feel has greatly improved this manuscript. We have provided a point-by-point response to all the comments below.

Reviewer #1

1. The model is far from the data and would be better if it focused on two classes of granules rather than a histamine stimulated GEF that is worth looking for, but premature for a model.

We have amended the model accordingly (Figure 11).

2. Please provide accession number for Rab45 and Rab46 sequences (upon first mention) so readers can find them in gene expression datasets

We have added the accession numbers for Rab46 (cracr2a-a) and cracr2a-c in the third paragraph of the introduction. We have not cited Rab45 in this manuscript.

3. GTPase activity is not what is shown as being needed here (Page 2 and page 4 header and page 7 text); mutant data are consistent with GTP binding being needed (as correctly stated on page 4 body of paragraph). Please correct throughout.

Please see amendments throughout text.

4. Page 7 first paragraph starts with: Histamine evokes nucleotide activation of Rab46. This has not yet been shown!

Amended in text.

Reviewer #2

1. The introduction is poorly written. There is no introduction of Weibel-Palade bodies

We have updated our introduction and introduced WPBs in the first paragraph.

2. It is incumbent upon the authors to provide the experimental evidence that convinces the reader that indeed this protein is a Rab, even if atypical.

We apologize for the confusion. In the results section under the heading 'Nucleotide binding is necessary for Rab46-dependent trafficking' we have now referred to Srikanth et al 2016, where the ability of Rab46 to hydrolyse GTP has been demonstrated.

3. On pg. 2, the following line "Several studies have shown that calmodulin, when activated by Ca²⁺, activates Rab GEFs 10" is misleading.

We have amended the text in the third paragraph of the introduction to show ‘calmodulin, when activated by calcium, plays a role in Rab activity’ and have added further references.

Specific comments:

4. In figure 1 the authors use an antibody against CRACR2A-L in immunofluorescence studies but provide no evidence that it is specific.

In the results section under the heading ‘Rab46 is a novel Rab GTPase localised to WPBs’ we have now referred to Wilson et al. 2015, our previous paper, where western blotting data demonstrated that the anti-Rab46 antibody only recognises a band of approx. 95kDa that is reduced when we deplete the cells of Rab46 using specifically targeted siRNA. In this study we also demonstrate that the anti-Rab46 antibody recognizes overexpressed Rab46 (Figure 1e).

5. As is, only 1 siRNA is shown, the siRNAs are only moderately efficient, it is unclear which siRNA is used, only a single cell is shown.

We have amended text and figure legends to show that in all experiments Rab46 siRNA-1 is used unless otherwise stated. We have also included extra data in the supplementary (Suppl. Figure S1c) to validate that depletion of Rab46 using siRNA-2 also inhibits histamine-dependent perinuclear trafficking.

6. The immunoblot in suppl. Fig. 1 is poor, is the inefficient knockdown a consequence of a partially selective antibody or partially effective siRNAs. Moreover, in figure 3, the immunofluorescent signal is completely lost.

We have replaced the western blot example in suppl. Fig S1.

Please refer to example images of Rab46 immunofluorescence above. We do have some residual green fluorescence in cells transfected with Rab46 siRNA. In our manuscript we detail the methods used for noise subtraction applied equally to images for presentation without affecting the underlying values.

7. Due to the low quality of manuscript presentation it is difficult to fully understand the exact nature of the primary data for figure 2a and 2d. This data should be presented and better explained.

We apologize for this confusion and have made it clearer in the text (please see results section paragraph 2) and Figure 2 legend that we have quantified images immunostained for vWF and Rab46 (Figure 2a) or vWF alone (Figure 2b and f). We have also added a representative image (Figure 2e).

8. Figure 3 demonstrates that knockdown of CRACR2A-L blocks histamine-stimulated translocation of vWF (in the WPBs) to the perinuclear region. This is interesting but needs to be better controlled. Use both siRNAs, they have a different knockdown efficiency, do they give a different phenotype/strength of phenotype? A functional rescue with wild-type CRACR2A-L compared to mutations that presumably influence GTP status.

As above, we have repeated these experiments using Rab46 siRNA-2 (Suppl. Figure S1c) which targets a non-coding region.

Functional rescue experiments have not been performed because the overexpression of wild-type Rab46 in these cells tends to drive Rab46 into an active state. However, we have added data to show that histamine does not induce WPB translocation when cells overexpress a mutant that is deficient in GTP binding (N658I). Indicating that Rab46 GTP status is important (Figure 4b).

9. In figure 4a, (now Figure 7c) its not that Rab46 is not localized to WPBs containing P-selectin, it's that P-selectin and Rab46 are expressed in different cells. This makes little sense and is not consistent with the conclusions.

We have added extra data (Figure 7a, b) to show we observe endogenous P-selectin and Rab46 in the same cells. As shown, sometimes we also observe P-selectin and angpt2 in separate cells but the model still supports that histamine regulates Rab46-positive WPB trafficking that don't carry P-selectin.

10. I don't understand the relationship of the short isoform to the G-domain-containing isoform in figure 8, why is it discussed?

Rab46 is an unusual G protein in that it has EF-hand Ca^{2+} -sensing domains in addition to the Rab domain. Surprisingly, we show that histamine-stimulated perinuclear localisation of WPBs is independent of Ca^{2+} therefore we wondered what role the EF-hands played in WPB trafficking. We know that a shorter isoform of the Rab46 gene, that lacks the Rab domain but has functional EF-hands, has important roles in T-cells. Srikanth et al. (Nature Cell Biology, 2010) demonstrated that Ca^{2+} binding to the EF-hands of cracr2a-c evokes dissociation of proteins clustered at the plasma membrane. This gave us some insight into the role of the EF-hands in Rab46, therefore we hypothesized that Ca^{2+} binding to Rab46 could evoke dissociation of

WPBs clustered at the MTOC. We have amended the text in the results section, 1st paragraph, under the heading 'Intracellular calcium evokes Rab46-dependent WPB dispersal' to make this clearer.

11. More importantly, the authors state that thapsigargin induces dispersal of histamine-clustered WPBs. Presumably this is based on the small decrease in the perinuclear concentration of endogenous Rab46, and yet there is no corresponding significant increase in the peripheral pool (Fig. 8b).

There is a small but significant decrease in the perinuclear concentration of Rab46 which was expected from a highly localised Ca²⁺-response. The graph (now Fig 10c and d) shows that upon thapsigargin treatment both endogenous and overexpressed wt Rab46 was dispersed into both the intermediate and the perinuclear regions.

12. The model in figure 9 indicates a direct interaction between dynein (not labeled in the legend) and the Rab-like domain of CRACR2A-L. This is readily testable.

We have added new data to show that Rab46 interacts with a dynein complex (Figure 6).

Reviewer #3

1. Previously accumulation of Weibel-Palade bodies at the MTOC has been reported by Vischer and co-workers (Vischer et al. 2000) as well as Rondaij and co-workers (Rondaij et al, 2006). Interestingly, in these studies clustering of Weibel-Palade bodies was induced by the administration of agonists like epinephrine and forskolin which raise intracellular cAMP-levels. In the current study clustering of Weibel-Palade bodies at the MTOC was induced by histamine. Inclusion of stimulation-experiments employing epinephrine or forskolin is needed to established whether Rab46 only promotes histamine-induced clustering.

Responses to these comments are reported below.

-In Figure 3 the authors show that siRNA-mediated down-regulation of Rab46 prevent histamine-induced clustering of Weibel-Palade bodies. In contrast, they report that thrombin does not induce clustering. This is surprising since thrombin has also been reported to increase intracellular Ca²⁺ levels thereby potentially activating Rab46 dependent clustering of Weibel-Palade bodies. As mentioned earlier cAMP-dependent clustering of Weibel-Palade bodies has been previously reported. It is crucial to address whether epinephrine or forskolin-induced clustering is also impaired in endothelial cells that lack Rab46. Inclusion of stimulation-experiments employing epinephrine or forskolin is needed to established whether Rab46 only promotes histamine-induced clustering.

-Clustering of Weibel-Palade bodies at the MTOC occurs in response to agonists like epinephrine and forskolin. These agents raise intracellular cAMP levels. In view of these

earlier data it would be useful to also explore whether Rab46 promotes cAMP dependent clustering of Weibel-Palade bodies.

Although both thrombin and histamine evoke increases in intracellular calcium, histamine-induced trafficking of WPBs to the MTOC is independent of calcium. Calcium is necessary for the dispersal of histamine-induced perinuclear localised WPBs.

Firstly, we have added qPCR and immunofluorescent imaging data to suggest that histamine induced clustering is via the H₁ receptor, which is a Gq-coupled GPCR (Figure 3c and 3d). In addition, we have added data to show that histamine induced perinuclear clustering is not inhibited in the presence of H-89, a broad PKA inhibitor, the downstream target of cAMP (Suppl. Figure 5a and 5b). These data suggest that cAMP is not necessary for histamine-induced perinuclear trafficking. Secondly, to see if Rab46 is necessary for stimulants that evoke cAMP-dependent perinuclear clustering, we have added images to show that, although epinephrine (with IBMX) does not induce robust WPB clustering at this acute time point, clustering is still observed in the absence of Rab46 expression (Suppl. Figure 5c). These data suggest that Rab46 is not necessary for cAMP-mediated clustering.

2. The claim that Rab46 couples physiological stimuli to selective release of cargo from Weibel-Palade bodies is not sufficiently supported by the data. Currently, this is only based on the presence of P-selectin and angiotensin-2 in distinct subset of Weibel-Palade bodies. As shown by van Aghtmaal and co-workers (van Aghtmaal et al., 2012) this is probably based on a staining-artefact. These must preferably include immuno-EM stainings for P-selectin and angiotensin-2 as well as release studies showing that following histamine stimulation only P-selectin and not angiotensin-2 is released. Control stimulation employing thrombin are needed to substantiate that under these conditions both P-selectin and angiotensin-2 are released

We describe our new angpt2 release experiments in the specific comments below. Re thrombin: In this study we first identify Rab46 as a G protein that mediates differential WPB body trafficking when acutely activated by histamine. Next, our emphasis was on elucidating the molecular and functional mechanisms underlying Rab46-dependent trafficking. The study does not investigate any mechanisms by which thrombin regulates WPB trafficking or cargo release (thrombin and histamine have previously been shown to elicit differential effects on WPB release mechanisms at the cell surface – please see Nightingale et al. J Thromb Haemost 2018), only to demonstrate that thrombin does not activate Rab46. Therefore, thrombin is not a direct comparator and we used PMA as a positive control in our release studies (see below). We have made the objectives of this study clearer by amending the title to

include 'inflammatory-coupled' not 'physiological' and have amended the summary and abstract to emphasise the mechanisms underpinning Rab46-dependent trafficking of WPBs.

- The data reported in Figure 4 need to be improved. First of all conflicting data have been reported with respect to the co-localisation of P-selectin and angiotensin-2 in Weibel-Palade bodies. An early study (Fiedler et al., 2004) showed that P-selectin and angiotensin-2 are stored in a mutually exclusive manner in Weibel-Palade bodies. In a later study, clear co-localisation of angiotensin-2 and P-selectin was observed (Van Agtmaal et al., 2012). First of all the authors need to show co-stainings of angiotensin-2 and P-selectin.

We have now moved our data demonstrating co-staining of P-selectin and angpt-2 in distinct WPBs using high-resolution imaging from supplementary to the main text (Figure 8a). We also demonstrate that in addition to P-selectin being localised to WPBs that are distinct from angpt-2-positive WPBs, high-resolution immunofluorescent images depict P-selectin localised to WPBs that are distinct from Rab46-positive WPBs (Figure 7a, b) corroborating the existence of these subsets of WPBs. We have used high-resolution AiryScan imaging, that has double the resolution of confocal microscopy (Van Agtmaal et al. 2012) and we have validated the specificity of our antibodies. Differences from the Van Agtmaal study may also be explained by their use of endothelial cells derived from peripheral blood progenitor cells, which have many of the characteristics of endothelial cells but there is some controversy regarding their phenotypic identity.

- The analysis shown in Figure 4A reveals that under the specific conditions of this experiment only a very limited of cells express P-selectin. P-selectin expression can be upregulated by pre-incubating with IL-4 and this would provide a means to obtain a more homogenous expression of P-selectin endothelial cells.

Figure 7b (was Figure 4a) is a representative image of P-selectin where we can also show that histamine induces clustering in neighbouring cells that only express Rab46. We have now added Figure 7a and e that show P-selectin expression is robust. In this study, we are observing a physiological response to acute histamine signals and pre-incubation with IL-4 may represent another physiological state.

- Also ultrastructural studies employing immune electron microscopy are needed to show that P-selectin and Rab46 do not co-localize on WPBs. In addition to histamine and PMA induced P-selectin expression also histamine and PMA-induced release of angiotensin-2 needs to be measured in cells treated with Rab46 siRNA.

Please see above re imaging. We have now added new functional data on histamine and PMA induced angpt2 release (Figure 8g). At a low 0.3 μ M dose of histamine (where there is a histamine-induced calcium response, Figure 9a, but no WPB

clustering at the MTOC, Suppl. Figure S4) we observe a small release of angpt2. Release cannot be potentiated with an increased physiological dose (30 μ M) of histamine (where we observe an optimal calcium response, Figure 9a, and WPBs cluster at the MTOC) suggesting WPB clustering inhibits further release. We know that WPB clustering is dependent on Rab46 so we were surprised that knock down of Rab46 decreased secretion. However, as we also observed reduced angpt2 in lysates from histamine stimulated cells transfected with Rab46 siRNA (not shown: indicating an increase in release) we explored the effect of depletion of Rab46 on angpt2 protein levels. Rab46 depletion does not affect angpt2 mRNA and but has a tendency to reduce protein expression. Our new data indicates that Rab46 could play a role in angpt2 recruitment (Figure 8j).

- In figure 4E quite some difference is observed between the P-selectin intensity of histamine and non-treated cells. The observed difference is not significant; larger numbers of cells will need to be analysed in order to establish whether P-selectin containing Weibel-Palade bodies are not trafficking to the MTOC.

We have added two more biological replicates to the data now depicted in graph Figure 8e. We have analysed five biological replicates, each replicate has a minimum of five random fields of view, each image contains approx. 10 cells. As per previous comment we have robust expression of P-selectin in our cells and we have also added new data to demonstrate P-selectin doesn't cluster with the Q604L constitutively active Rab46 (Figure 7e).

3. In the last part of the manuscript over-expression of Rab46 variants in endothelial cells is being employed. As can be seen from the images shown in Figure 6-8; the number of Weibel-Palade bodies is greatly reduced in transfected cells. This limits the conclusions that

can be derived from these experiments. Preferably, viral systems providing a more homogenous expression should be used for these experiments. Shown is a panel of images to verify

homogenous expression of our mutants. HUVECs transfected with Rab46 mutants (inactive N658I and active Q604L) and immunostained with anti-Rab46 antibody 24 hours post-transfection show homogenous expression of both mutants.

Fig. 4a (was Figure 6) demonstrates the effects of nucleotide-binding mutations on Rab46 distribution. We have added new data to show vWF localisation with wild type overexpressed Rab46 (Figure 1e) and the effects of histamine on WPB distribution in cells expressing the GTP-binding mutant (N658I) (Figure 4b), WPBs are evident in these cells.

Re: the EF-hand previously in Figures 7 and 8 (now Figure 9 and 10) please refer to the specific responses to comments 12 and 13 below.

4. The authors show that Rab46 is present on Weibel-Palade bodies. Based on their high-resolution imaging it appears that Rab46 is not evenly distributed on Weibel-Palade bodies. The images shown in Figure 1a reveals that Rab46 is also localized on another subcellular compartment. Did the authors attempt to define to which other subcellular compartment Rab46 is localized?

To ensure that Rab46 is not localised to other vesicles we have also stained for lamp1 (lysosomes,) tPA granules and added new data for Rab11 (recycling vesicles) (Figure 1d). In addition, overexpressed wild type Rab46 is only localised to WPBs (Figure 1e).

5. In Figure 2 the authors show that Rab46 is only present on around 50% of Weibel-Palade bodies. This observation is based on co-localization studies; additional information is needed to show that only a subset of Weibel-Palade bodies contain Rab46; electron microscopy studies would probably be useful.

Our localisation studies are based on AiryScan imaging (Zeiss) which is high-resolution imaging offering double the resolution of confocal. Please refer to Figure 1c where we can observe Rab46 localised to individual WPBs. We agree that going forward it would be very interesting to determine the Rab46/WPB ratio using EM.

6. Also this part of the study needs to be complemented with real-time analysis of Weibel-Palade body trafficking to demonstrate whether only Rab46-positive Weibel-Palade can traffick to the MTOC.

Aside from determining the effect of conserved mutations in the Rab protein domains on Rab46 distribution, this study is based on endogenous Rab46. We have found that overexpression of Rab46 in endothelial cells disturbs Rab46 kinetics, driving Rab46 into an active state. We have also shown and discussed that GFP tagging of the N-terminus hinders the EF-hand so that Rab46 cannot be released from microtubules at the MTOC. However, we are generating a new tagging system with the aim of developing CRISPr cells/mice for future studies.

7. It would also be needed to show how Weibel-Palade bodies acquire Rab46. Does only a subset of Weibel-Palade bodies acquire Rab46? How is this regulated? Which effectors mediate Rab46-dependent trafficking to the microtubule organizing center?

We agree these are all interesting questions, but to describe all the GAPs, GEFs and effector proteins is beyond scope of this study and will form part of our research for the next 5 years. However, we have added new mechanistic data to show that the dynein forms part of a Rab46 effector complex (Figure 6), agreeing with a recent publication from Wang et al. 2019. In addition, we have added data that suggest Rab46 may play a role in the recruitment of angiopoietin-2 (Figure 8j).

8. In figure 2 the authors report that siRNA-mediated down-regulation of Rab46 affects the VWF content in endothelial cells. They also report an increase in the number of Weibel-Palade bodies under these conditions. Did the authors investigate whether changes in the multimer pattern of VWF occur under these conditions? It is well-established that the propensity of VWF to form polymers is highly dependent on its ability to be stored in Weibel-Palade bodies.

This is a very interesting question but is beyond the scope of this mechanistic study of Rab46.

9. Data contained in Figure 6 are of interest. Results are obtained employing transient expression of Rab46 variants. Therefore only a limited number of cells are expressing the Rab46 variants. Using these methods only morphological analyses are feasible. The confocal images displayed in this figure need to be extended to also show whether Weibel-Palade bodies are present and whether they are localized to the MTOC or in the periphery of the cell. Stainings should preferably be performed for VWF, P-selectin and angiopoietin-2; also the MTOC needs to be visualized in transfected cells.

Fig. 4a (was Figure 6) demonstrates the effects of nucleotide-binding mutations on Rab46 distribution to further characterise Rab46 as a GTPase (i.e. mutation of well conserved nucleotide binding affects the distribution of the Rab). To verify the role of Rab46 GTP activity on WPB trafficking we have added data to show that when GTP cannot bind to Rab46 (N658I) then histamine does not induce clustering of Rab46 or WPBs (vWF: Figure 4b). In addition, we have added images to show that when Rab46 is constitutively (Q604L) active then it is co-localised with WPBs (vWF) at the MTOC (pericentrin: Suppl. Figure S3b). We have also added new data to show that Rab46 Q604L localises to the MTOC but this is distinct from P-selectin (Figure 7e), however Rab46 Q604L localises angpt2 to the MTOC (pericentrin: Figure 8f).

10. Ideally viral delivery systems (adenovirus & lentivirus have been developed for endothelial cells) would allow for performing quantitative biochemical assays to monitor the effect of the Rab46 variants on Weibel-Palade body release and clustering.

Please see response to comment 3.

11. In figure 7 AM-BAPTA is being used to interrogate whether intracellular Ca²⁺ is needed for clustering of Weibel-Palade bodies. Following incubation with AM_BAPTA Rab46 is present at the MTOC; it is not clear from the images shown in Figure 7e whether Weibel-Palade bodies also cluster at the MTOC in the presence of AM-BAPTA since the cells in which Rab46 localizes to the MTOC contain a very limited number of Weibel-Palade bodies.

Incubation with AM-BAPTA is to show that clustering at the MTOC is calcium independent. We have split the channels in the images shown in Figure 9e (formerly Figure 7e) into the representative channels to show that both Rab46 and vWF are localised to a perinuclear area upon histamine stimulation in the absence or presence of BAPTA.

12. Also in Figure 7 cells that express Rab46-EFmut display very limited number of Weibel-Palade bodies. The lack of sufficient number of Weibel-Palade bodies in transfected cells does not allow for monitoring of the effect of the Rab46-EF mutant on Weibel-Palade body clustering.

In Figure 9 (previously 7) we show that histamine-induced clustering of endogenous Rab46 and WPBs (vWF) is calcium independent by using a pharmacological inhibitor of calcium (Figure 9e: histamine evoked trafficking still occurs in the presence of BAPTA). We know this trafficking is dependent on Rab46 (Figure 3) so to understand the effect of calcium on Rab46 dynamics we over express the EF mutant. The EF mut is a tool to see if a Rab46 protein that is unable to bind to calcium is still able to localise to the MTOC.

If we observe early expression of the EF hand mutant (as shown in these two examples) we can see that it localises to WPBs (vWF: red) before orientating towards the MTOC, suggesting that it traffics like the wild-type protein, until it becomes lodged at the MTOC. At this earlier time point you can also observe WPBs (vWF) which have also orientated to the MTOC. However, we agree that long term overexpression of this mutant has an effect on the WPB population which could be due to degradation. However, here we emphasise that data from Figure 9 and Figure 10 are for deciphering the mechanisms underlying Rab46 function, we have amended the text in the final paragraph of the results to state 'Thapsigargin induced the dispersal of wt Rab46 but not the EF-hand mutant suggesting that it is Ca²⁺ binding to the EF-hand of Rab46 that is necessary for dispersal of Rab46 from the perinuclear clusters'. From this data we can infer that, as the Rab46 EF mutant is able to localise to the MTOC then, Rab46-dependent trafficking to the MTOC is calcium independent.

13. In Figure 8 primary data are not shown for cells expressing WT-Rab46 and EF-Hand mutant. The effects of thapsigargin treatment as shown in Figure 8c are very small. The data indicate that at best thapsigargin only very limitedly affects Rab46 localization in histamine-treated cells. In addition a marker for Weibel-Palade bodies needs to be added to these images in order to monitor the effect of thapsigargin, WT-Rab46 and EF-Hand mutant on the number and localization of Weibel-Palade bodies.

We now show example images of thapsigargin treatment of wt and EF hand Rab46 expression (Figure 10). We expected the effects of thapsigargin to be small (but significant) as although thapsigargin induces calcium release from intracellular stores only the calcium released from a localised nanodomain (at the MTOC) will be effective.

The physiological effect of intracellular calcium release is shown in Figure 10b where release of calcium from intracellular stores induces dispersal of histamine-evoked clusters of endogenous Rab46 and vWF from the MTOC. To understand if the calcium binding properties of Rab46 plays a role in this trafficking we again have used the EF hand mutant as a tool. The emphasis here is on mechanisms that underly Rab46 function. Compared to wt and endogenous Rab46, the EF hand mutant is unable to disconnect from the MTOC, even in the presence of calcium. Although we have no WPBs in some of these cells, this allows us to infer that calcium has to bind to the EF-hand to release Rab46 from the MTOC.

April 11, 2019

RE: JCB Manuscript #201810118R

Dr. Lynn Mckeown
University of Leeds
LICAMM
Clarendon Way
Leeds LS2 9JT
United Kingdom

Dear Dr. Mckeown,

Thank you for submitting your revised manuscript entitled "A Ca²⁺-regulated G-protein (Rab46) couples inflammatory stimuli to differential trafficking of Weibel-Palade bodies". Thank you for your efforts to address the concerns raised in peer review. You will see that both returning reviewers find that the revision is now stronger and recommend publication of the study in JCB without additional work. We would be happy to publish your paper in JCB pending final revisions necessary to meet our formatting guidelines (see details below).

1) Text limits: Character count for Articles and Tools is < 40,000, not including spaces. Count includes title page, abstract, introduction, results, discussion, acknowledgments, and figure legends. Count does not include materials and methods, references, tables, or supplemental legends.

2) Titles, eTOC: Please consider the following revision suggestions aimed at increasing the accessibility of the work for a broad audience and non-experts.

Title suggestions:

Rab46 integrates Ca²⁺ inputs and histamine signaling to regulate selective cargo release from Weibel-Palade bodies

Running title (50 characters max, including spaces): Rab46 regulates selective Weibel-Palade body release (word count slightly closer to our limit, we should be able to accommodate a slight extension)

eTOC summary: A 40-word summary that describes the context and significance of the findings for a general readership should be included on the title page. The statement should be written in the present tense and refer to the work in the third person.

Suggested eTOC with edits to meet JCB style:

It is unclear how a plethora of stimuli evoke differential cargo secretion from endothelial cells in order to produce stimulus-appropriate responses. Miteva, Pedicini et al. show that Rab46 integrates histamine signaling and Ca²⁺ signals to regulate selective cargo release from Weibel-Palade bodies.

3) Figure formatting:

- Scale bars must be present on all microscopy images, including inset magnifications. Please add scale bars to 1b (magnification), 7b, 10de
- Molecular weight or nucleic acid size markers must be included on all gel electrophoresis. Please add molecular weight with unit labels on the following panel: S1a
- Please note that all blots should include unit labels (e.g., 6bc)

4) Statistical analysis: Error bars on graphic representations of numerical data must be clearly described in the figure legend. The number of independent data points (n) represented in a graph must be indicated in the legend. Statistical methods should be explained in full in the materials and methods. For figures presenting pooled data the statistical measure should be defined in the figure legends.

Please indicate n/sample size/how many experiments the data are representative of: 9cd

5) Materials and methods: Should be comprehensive and not simply reference a previous publication for details on how an experiment was performed. Please provide full descriptions in the text for readers who may not have access to referenced manuscripts.

- Please include all siRNA oligo sequences if available to you from the manufacturer (including controls).
- Microscope image acquisition: The following information must be provided about the acquisition and processing of images:
 - a. Make and model of microscope
 - b. Type, magnification, and numerical aperture of the objective lenses
 - c. Temperature
 - d. imaging medium
 - e. Fluorochromes
 - f. Camera make and model
 - g. Acquisition software
 - h. Any software used for image processing subsequent to data acquisition. Please include details and types of operations involved (e.g., type of deconvolution, 3D reconstitutions, surface or volume rendering, gamma adjustments, etc.).

6) References: There is no limit to the number of references cited in a manuscript. References should be cited parenthetically in the text by author and year of publication. Abbreviate the names of journals according to PubMed.

- Please be sure to format references per JCB style as described above.
- Please also be sure to format preprint references as per JCB guidelines:
<http://jcb.rupress.org/reference-guidelines>

7) A summary paragraph of all supplemental material should appear at the end of the Materials and methods section.

8) Thank you for providing source code. Generally, these are provided as raw code in a .txt file or as other file types in a .zip file. Please also include a one-sentence summary of each file in the Online Supplemental Material paragraph of your manuscript.

A. MANUSCRIPT ORGANIZATION AND FORMATTING:

Full guidelines are available on our Instructions for Authors page, <http://jcb.rupress.org/submission-guidelines#revised>. **Submission of a paper that does not conform to JCB guidelines will delay the

acceptance of your manuscript.**

B. FINAL FILES:

-- High-resolution figure and video files: See our detailed guidelines for preparing your production-ready images, <http://jcb.rupress.org/fig-vid-guidelines>.

Thank you for this interesting contribution, we look forward to publishing your paper in the Journal of Cell Biology.

Sincerely,

Pier Paolo Di Fiore, MD, PhD
Editor, Journal of Cell Biology

Melina Casadio, PhD
Senior Scientific Editor, Journal of Cell Biology

Reviewer #2 (Comments to the Authors (Required)):

The revised manuscript addresses my issues.

Reviewer #3 (Comments to the Authors (Required)):

My comments have been dealt with adequately